# Neural and computational mechanisms of momentary fatigue and persistence in effort-based choice

Tanja Müller [1,2 ✉], Miriam C. Klein-Flügge [1,2], Sanjay G. Manohar [2,3], Masud Husain[1,2,3,6] & Matthew A. J. Apps [1,2,4,5,6 ✉]

From a gym workout, to deciding whether to persevere at work, many activities require us to persist in deciding that rewards are 'worth the effort' even as we become fatigued. However, studies examining effort-based decisions typically assume that the willingness to work is static. Here, we use computational modelling on two effort-based tasks, one behavioural and one during fMRI. We show that two hidden states of fatigue fluctuate on a moment-to-moment basis on different timescales but both reduce the willingness to exert effort for reward. The value of one state increases after effort but is 'recoverable' by rests, whereas a second 'unrecoverable' state gradually increases with work. The BOLD response in separate medial and lateral frontal sub-regions covaried with these states when making effort-based decisions, while a distinct fronto-striatal system integrated fatigue with value. These results provide a computational framework for understanding the brain mechanisms of persistence and momentary fatigue.

[1] Department of Experimental Psychology, University of Oxford, Oxford, UK. [2] Wellcome Centre for Integrative Neuroimaging, University of Oxford, Oxford, UK. [3] Nuffield Department of Clinical Neurosciences, University of Oxford, Oxford, UK. [4] Centre for Human Brain Health, School of Psychology, University of Birmingham, Birmingham, UK. [5] Institute for Mental Health, School of Psychology, University of Birmingham, Birmingham, UK. [6] These authors contributed equally: Masud Husain, Matthew A. J. Apps. ✉email: tanja.mueller@psy.ox.ac.uk; M.A.J.Apps@bham.ac.uk

Most daily tasks require the exertion of effort over an extended period of time. From a workout at the gym to deciding whether to persist with a task at work, much of our activities require us to keep deciding whether the effort is 'worth it'. However, declines in our willingness to work often co-occur with sensations of 'fatigue'. Such sensations are a common debilitating symptom across many psychiatric and neurological disorders and have dramatic impacts on levels of daily activity[1–3]. Research has begun to provide a richer understanding of the computational and neural mechanisms underlying how people value, and decide, whether a given amount of effort is 'worth it' for a certain magnitude of reward[4–12]. However, implicitly such studies have not accounted for the effects of fatigue or have attempted to control for any of its potential effects. Yet, the willingness to work is not static[13]. Sometimes even though the objective difficulty of a task remains the same individuals give up or take a break[14–18].

What are the hidden internal states that change how we subjectively value effort over time and prevent us from persisting? Theories suggest that the willingness to work can be characterised by cost-benefit trade-offs, where the value of a reward is subjectively discounted by the effort required to obtain it. Theoretically, we are willing to work when we consider the value of a reward worth the effort we have to exert to obtain it[13,19]. Although a number of factors can influence such valuations, it has been argued that sensations of fatigue induced by the exertion of effort can lead to subsequent reductions in the willingness to work. Theoretically, as fatigue intensifies, it increases the devaluation of rewards by effort, leading to reductions in the willingness to persist with a task, with less rewarding and more effortful acts likely to be avoided[13,20–22]. In addition, time spent resting can have a restorative effect[18], reducing fatigue and concomitantly increasing the willingness to exert effort to obtain rewards. Despite these claims being at the cornerstone of theoretical accounts, few studies have directly tested these tenets. Existing research has shown that higher levels of fatigue are related to a reduced willingness to exert effort for rewards[18,23]. But little work has examined the dynamic, moment-to-moment changes in how willing we are to decide that a reward is worth it for the effort[13,19–21,24].

Moreover, separate lines of evidence suggest that fatigue may be comprised of distinct components that operate on different timescales. There are short-term increases in fatigue during tasks, which can be reduced by short periods of rest (recoverable fatigue (RF))[14]. In addition, there are also longer-term changes that occur after extended periods of exertion for which simply resting may not lead to restoration (unrecoverable)[25]. As they increase, both components putatively also increase the devaluation of rewards by effort. However, although these components have been examined separately, there has yet to be a formal framework that unifies them, quantifies dynamic changes in fatigue, and how fatigue shifts the value people attribute to exerting effort to obtain rewards on a moment-to-moment basis.

Previous neurophysiological and neuroimaging accounts have highlighted a core system in the brain that processes the costs and benefits of engaging in effortful activities. Activity in sub-regions of the supplementary motor area (SMA)/anterior cingulate cortex (ACC), the middle frontal gyri (MFG), frontal pole (FP), and ventral striatum (VS) have been implicated in computing value and motivating the exertion of effort[4,5,7,26–32]. Evidence suggests that these regions also change their response with time on task[13,15,17]. However, it is unclear how this distributed system changes on a moment-to-moment basis, and how that might lead to changes in the value ascribed to exerting effort for reward. Do separate sub-regions within this network signal fatigue on different timescales and integrate this into computations of value?

Here, we hypothesise that the brain regions outlined above, that have previously been linked to effort-based decision-making, covary differentially with RF, unrecoverable fatigue (UF), and the momentary value of working (fatigue-weighted value). To test this notion, we designed an effort-based decision-making task (Fig. 1) where participants had to exert physical effort (grip force) to obtain rewards (credits) whilst undergoing functional Magnetic Resonance Imaging (fMRI). On each trial of this task, they chose between a 5 s 'rest' (no effort, low reward [1 credit]) and 5 s of 'work' which varied in effort (30–48% of maximum grip strength) and reward (6–10 credits). Using a computational model combining previous cost-benefit valuation models with latent fatigue variables, we predicted how effort-based decisions would change across the task, as well as in a separate behavioural study where people rated their momentary fatigue (Supplementary Materials). Using this design, we were then able to identify and dissociate brain regions in which the blood oxygen level dependent (BOLD) signal varied with the hidden variables of RF, UF, and fatigue-weighted subjective value (SV) on a trial-by-trial basis at the time of making effort-based decisions.

We find distinct contributions of the MFG, and two distinct sub-regions of the medial frontal cortex, in which activity covaries with the hidden 'fatigue' states that modulate the willingness to exert effort for reward. In addition, we distinguish these regions from a fronto-striatal system that integrates these hidden states with reward and effort to reflect the current value of working. The same computational model could also explain trial-to-trial ratings of fatigue, highlighting that the effects on effort-based decisions may be directly linked to sensations of fatigue. Our approach dissects hidden variables that underlie moment-to-moment changes in fatigue and effort-based decisions, providing a framework for examining how fatigue impacts behaviour in health and disease.

## Results

The aim of this study was to examine how the value attributed to exerting effort to obtain rewards changes on a moment-to-moment basis, due to fluctuations in internal states putatively linked to fatigue. We designed an effort-based decision-making task in which participants would choose between 5 s of 'work' or 5 s of 'rest' whilst undergoing fMRI (Fig. 1; Supplementary Fig. 1). Rest required no exertion, but only resulted in accruing a small reward (1 credit). The work offer varied on every trial in both reward magnitude (6, 8, 10 credits), as well as the amount of physical effort (grip force) required to obtain it. These effort levels were calibrated to participants' own maximum grip strength (30, 39, 48% of maximum voluntary contraction [MVC]). Participants had to exert force at the required level for a total of 3 s out of the 5 s window in order to receive the reward. Failure to do so resulted in 0 credits. Although these effort levels were demanding, they were easily achievable, and participants were successful at executing the required levels of force on over $M > 96.8\%$ (SD < 4.6) of trials at all levels of grip force. In addition, we required participants to rate their level of 'tiredness'—a more commonly used synonym of fatigue—between 0 and 10 before the start of the main task, and then again after completion of the experiment. Although participants could freely choose to rest, and thus prevent a significant build-up of fatigue, a repeated measures $t$-test revealed that ratings of fatigue were higher at the end of the experiment than at the beginning ($t(34) = 4.27$, two-tailed $p < 0.001$, Cohen's $d = 0.72$, 95% CI = [0.54, 1.52]; Supplementary Fig. 2).

**Effort discounting and persistence depend on the history of effort exerted**. We hypothesised that people's willingness to exert

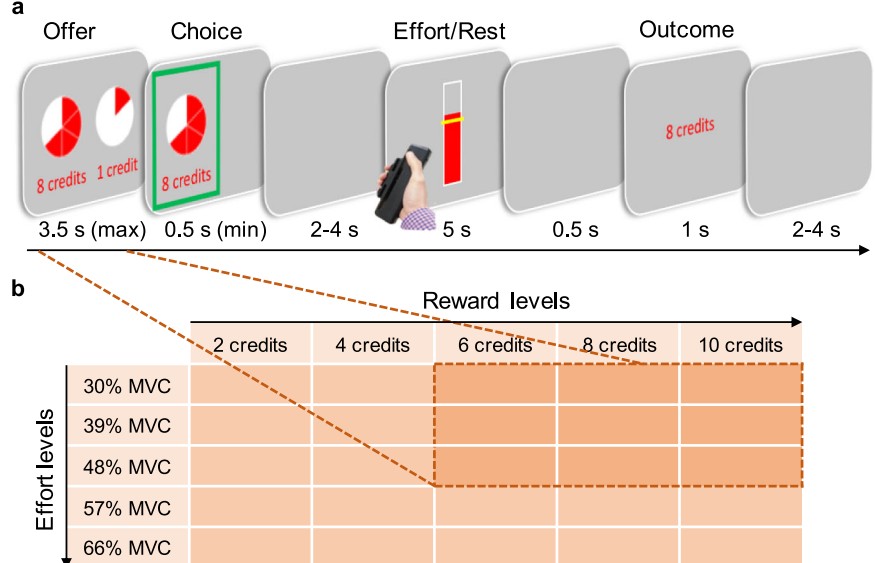

**Fig. 1 Trial structure and experimental design. a** Participants made choices between a work offer and a rest, over 216 trials. Rest was always worth a low reward (1 credit), with the work offer varying in reward (6, 8, 10 credits) and effort (30, 39, 48% of maximum voluntary contraction [MVC] on a hand-held dynamometer). Each participant's MVC was obtained prior to the experiment, in order to set the force levels idiosyncratically for each participant based on their grip strength. Effort levels were depicted as the number of elements in a pie chart—more elements signified higher effort. Participants performed a training session to familiarise themselves with how much force was required at each level of effort and to associate those effort levels with the corresponding pie charts prior to entering the scanner. The location (left/right) of the work offer and rest was randomised across trials. After making a choice, it was highlighted by a green frame. Participants then either rested or exerted the effort that was offered for 5 s. To obtain the credits offered, participants had to exert the required force, indicated by a yellow line, for a sum total of 3 s out of the 5 s window, with a red colouring providing online visual feedback. If unsuccessful they would receive 0 credits, if successful they would receive the credits of the offer that was chosen. The offer period was jittered independently of the other events allowing us to examine activity time-locked to effort-based decisions. In a pre-task, only a random subset of 10% of trials were selected and participants required to exert the effort (or rest) to obtain rewards. **b** Offers in the main and pre- tasks. Offers in the main task (dark orange) were restricted to those highest in value (higher reward, lower effort) to ensure that participants did not rest in any of the options in the main task simply because they would never value the options as worth working for. In a pre-task conducted outside the scanner, a wider range of offers (light orange) were included to ensure that each participant's effort-discounting behaviour could be quantified when they were not fatigued.

effort for reward would change across trials. More specifically, our computational model built around theories of fatigue suggests that exerting effort increases levels of fatigue, and when fatigue is higher, the same levels of effort and reward would have a reduced value compared to when fatigue is low. Such an account would predict that (i) participants would be more likely to work in situations where fatigue is low and (ii) people would shift their valuations, such that while higher effort/lower reward offers would be worth working for at some times, they would be avoided in favour of a rest at other times.

To test the first of these predictions we compared choices in the main part of the experiment, where every choice of work resulted in the requirement to exert force for reward, with a pre-task in which only a random 10% of trials resulted in the requirement to exert force (Fig. 1). The pre-task also contained a wider range of offers, including options that were lower in reward (2, 4, 6, 8, 10 credits), but higher in effort to ensure we could capture the full range of variability in people's tendency to discount rewards by effort (30, 39, 48, *57, 66% MVC*). This pre-task therefore allowed us to measure people's tendency to discount rewards by effort in a task where little fatigue would be accrued. First, we wanted to show that in this pre-task, participants were discounting rewards by effort. A logistic regression (with a Wilcoxon test across participants) on choices to work or rest showed evidence of this, with participants more likely to choose 'work' at higher rewards ($Z = 4.43$, $p < 0.001$, 95% CI = [1.15, 2.17]), but less likely to work at higher efforts ($Z = -5.11$, $p < 0.001$, 95% CI = [−2.55, −1.35]). However, despite showing effort-discounting effects, participants chose to work on almost all of the trials ($M \geq 95.4\%$) for each

combination of the higher reward (6–10 credits) and lower effort levels (30–48% MVC) that were included in the main task (Fig. 2). Thus, when little fatigue could accumulate, participants valued these offers consistently higher than the value of rest.

The second claim that would be predicted by our model is that the value of exerting effort for rewards declines as fatigue accumulates, i.e. participants would shift the value they ascribed to work offers across the main task. To test this, we performed a logistic regression on choices to work or rest (with a Wilcoxon test across participants), using effort and reward (offered on each trial), cumulative effort (sum total of effort exerted from all previous trials) and corresponding interactions as predictors (Fig. 2). Consistent with a dynamic change in the value of working, there was a significant interaction between effort and cumulative effort, as well as main effects of effort, reward and cumulative effort (cumulative effort × effort: $Z = -4.19$, $p < 0.001$, 95% CI = [−1.39, −0.47]; effort: −4.35, $p < 0.001$, 95% CI = [−1.83, −0.96]; reward: $Z = 3.96$, $p < 0.001$, 95% CI = [0.69, 1.71]; cumulative effort: $Z = -3.83$, $p < 0.001$, 95% CI = [−2.45, −0.84]). The three-way ($Z = -0.27$, $p = 0.79$, 95% CI = [−0.25, 0.37]) and the cumulative effort × reward interactions ($Z = 1.40$, $p = 0.16$, 95% CI = [−0.06, 0.42]) were not significant. Importantly, offers that were considered as higher in value and were chosen to work for at a high proportion in the pre-task, became gradually less and less likely to be selected across trials in the main task with effort and reward having strong effects even in the last 27 trials of the experiment (Fig. 2; Supplementary Fig. 2). Such findings are inconsistent with boredom or other factors leading to generally more noisy or random behaviour.

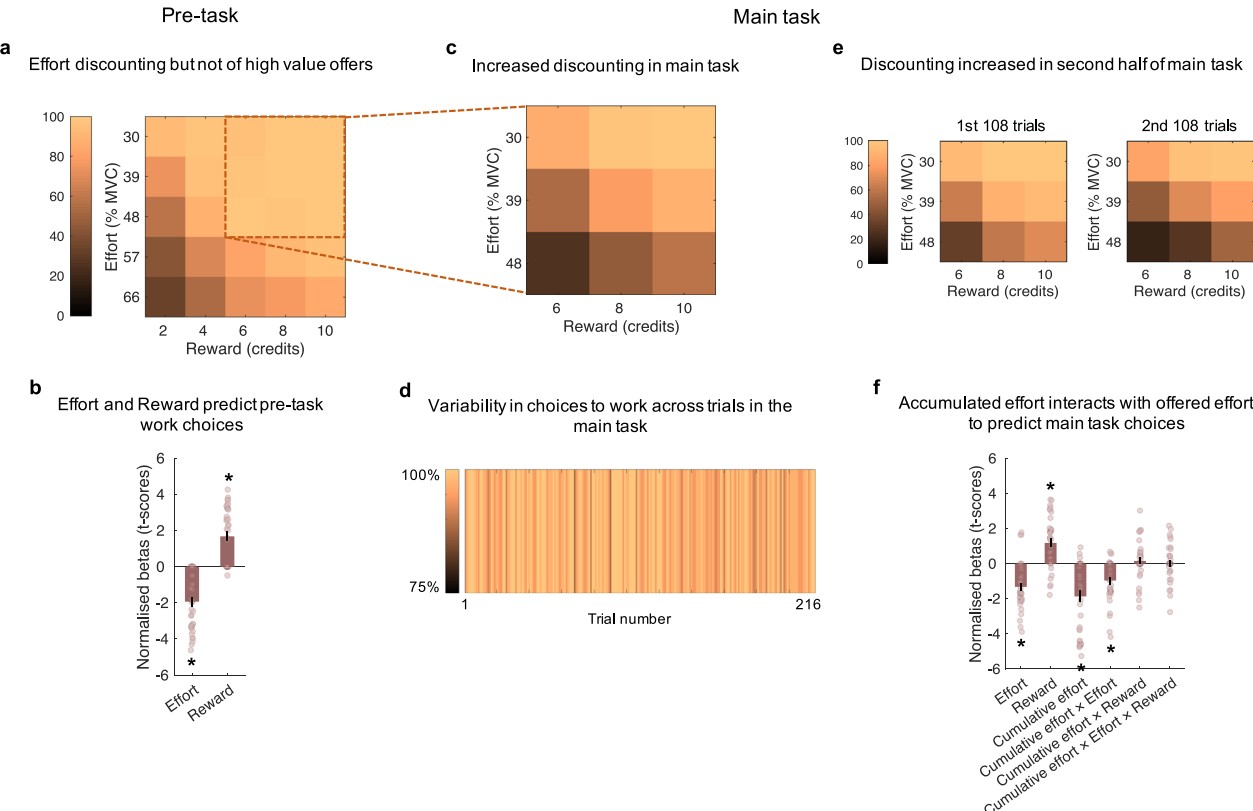

**Fig. 2 The shifting of subjective value of effort and reward. a** Proportions (means) of choices to work in the pre-task (where only 10% of trials resulted in subsequent work or rest). Participants were more likely to choose to work at higher reward and lower effort. The 'high value' work options (inside the dotted line) were almost always chosen as worth working for. **b** Logistic regression on the pre-task choice data shows significant positive effects of reward and negative effects of effort. The asterisks show significant $t$-scores (two-tailed $p < 0.001$) and the error bars represent SEM; $n = 36$. **c** Proportions (means) of choices to work in the main task, where all choices resulted in subsequent work or rest. Some of the high value options previously worth working for are now sometimes avoided. The lowest value of the higher value options (48% effort, 6 credits) is often avoided in favour of a rest in the main task. **d** Percentage of participants who accepted the work offer, illustrated separately for each trial in the main task. Values reflect the consistency with which trials were accepted or rejected across the experiment. This shows considerable variability in choices, but high levels of choices to work in early trials, rather than late. **e** Proportion (mean) of accepting the work offers in first and second half of the main task. Gradually increasing effort discounting reflects a shift in valuation of previously high valued offers across time in the main task. **f** Logistic regression predicting choices to work or rest in the main task. The effort level in the current trial's work offer interacts with the sum total of effort accumulated up to that trial (cumulative effort) to predict choice. The asterisks show significant $t$-scores (two-tailed $p < 0.001$) and the error bars represent SEM; $n = 36$.

Instead, these results are indicative of participants changing their subjective valuation of working across trials, with accumulated efforts increasing the discounting effect of effort and reducing the value of working across the experiment.

**Computational modelling: unrecoverable and recoverable fatigue states impact the subjective value of work.** To more precisely quantify changes in valuation of effort across the task, we developed a computational model of fatigue and its effect on effort-based decisions (Fig. 2; Supplementary Fig. 3). We hypothesised that fatigue has several components that impact on the willingness to exert effort, each of which operates on a different timescale. We predicted that the value of working fluctuates on a short-term basis due to a build-up of (recoverable) fatigue after exerting effort that is recovered by rest[14]. A separate line of evidence suggests that demanding tasks also cause (unrecoverable) 'executive' fatigue that cannot be easily restored just by taking some time resting[25]. Here, we formalised a computational model of how these two sources of fatigue would fluctuate trial-to-trial during the task (Fig. 3; see Methods). This model contained three free parameters estimated on choices to work or rest in the main task. One parameter ($\alpha$) scaled the amount that RF

increased by the exertion of effort, with a second ($\delta$) scaling the amount that RF was reduced by time spent resting. The third parameter scaled the amount that UF increased by exerting effort ($\theta$). The fluctuating recoverable, and gradually increasing unrecoverable, fatigue values were integrated into a parabolic effort-discounting model[4,33,34], in which rewards were devalued more by effort as levels of the UF and RF increased. To account for individual differences in people's effort-discounting when participants were not fatigued, we also included a free parameter ($k$) which scaled how much participants devalued rewards by effort, fitted to choices in the pre-task, and carried over as a fixed parameter to model main task choices.

To test whether shifts in people's valuations of effort across the main task were related to hidden recoverable and unrecoverable states, and whether three parameters were necessary to explain changes in the willingness to work, we performed model comparison. The results showed that the full model fitted better to participants' decisions to work than four other models in a factorial design (Fig. 3) as well as two further mathematically plausible models, in which fewer parameters scaled or removed the unrecoverable and recoverable components (Supplementary Table 5). Findings were comparable when using BIC or AIC suggesting that extra parameters in the full model increased

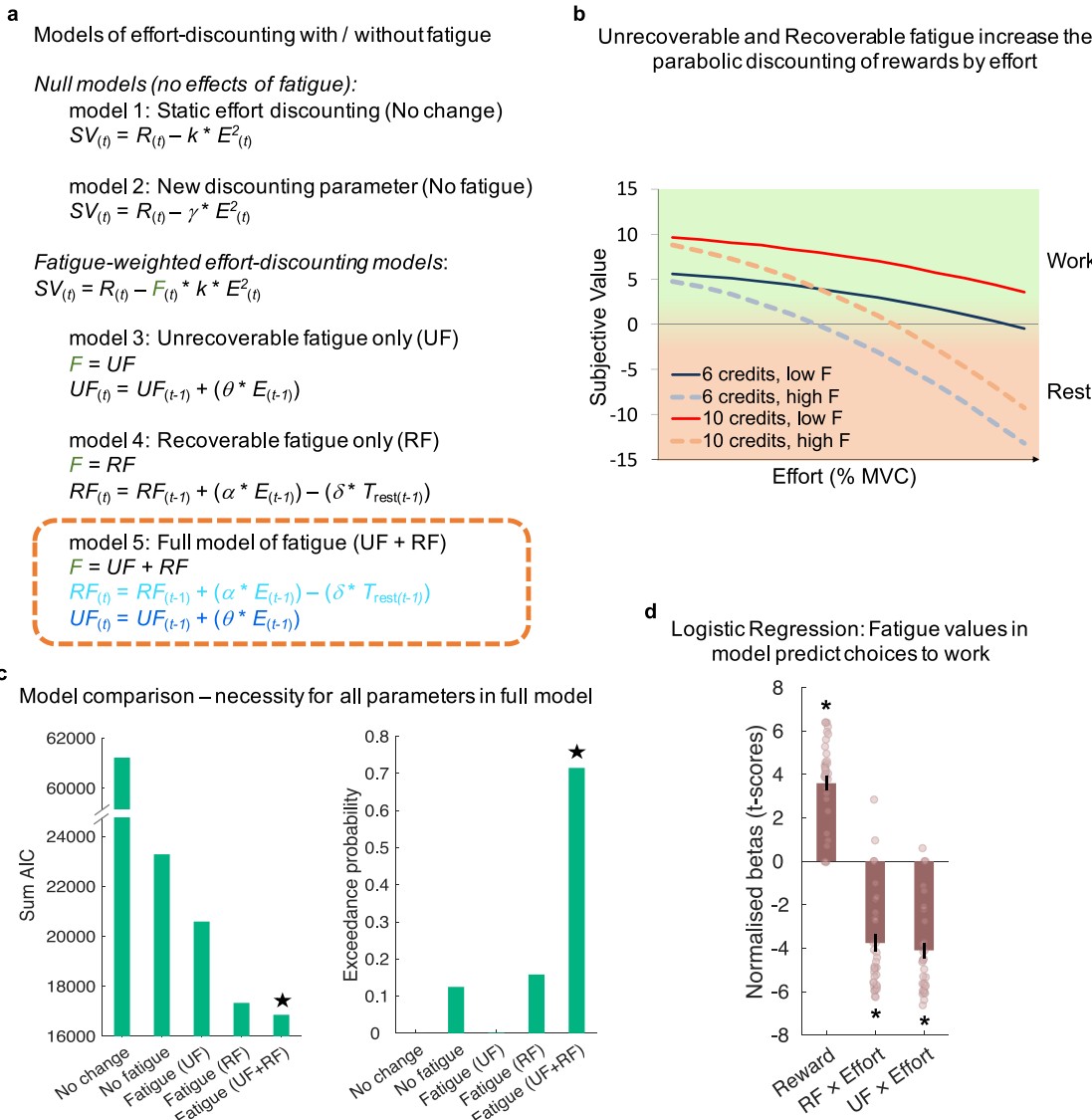

**a** Models of effort-discounting with / without fatigue

*Null models (no effects of fatigue):*
model 1: Static effort discounting (No change)
$SV_{(t)} = R_{(t)} - k * E^2_{(t)}$

model 2: New discounting parameter (No fatigue)
$SV_{(t)} = R_{(t)} - \gamma * E^2_{(t)}$

*Fatigue-weighted effort-discounting models:*
$SV_{(t)} = R_{(t)} - F_{(t)} * k * E^2_{(t)}$

model 3: Unrecoverable fatigue only (UF)
$F = UF$
$UF_{(t)} = UF_{(t-1)} + (\theta * E_{(t-1)})$

model 4: Recoverable fatigue only (RF)
$F = RF$
$RF_{(t)} = RF_{(t-1)} + (\alpha * E_{(t-1)}) - (\delta * T_{rest(t-1)})$

model 5: Full model of fatigue (UF + RF)
$F = UF + RF$
$RF_{(t)} = RF_{(t-1)} + (\alpha * E_{(t-1)}) - (\delta * T_{rest(t-1)})$
$UF_{(t)} = UF_{(t-1)} + (\theta * E_{(t-1)})$

**b** Unrecoverable and Recoverable fatigue increase the parabolic discounting of rewards by effort

**c** Model comparison – necessity for all parameters in full model

**d** Logistic Regression: Fatigue values in model predict choices to work

**Fig. 3 Modelling the fatigue-weighted subjective value of effort and reward. a** List of main models compared. All models assume that rewards ($R$) increase subjective value ($SV$), effort ($E$) decreases SV, and people discount the rewards by effort idiosyncratically—modelled with a discount parameter ($k$) fitted to the pre-task. Two null models assume that the willingness to exert effort for reward remains static across the trials of the main task, either with the same discounting parameter as for the pre-task ($k$; model 1) or with a new discounting parameter ($\gamma$; model 2). Models 3–5 capture changes in effort discounting due to fatigue. The full model (model 5) assumes that exerting effort increases recoverable fatigue ($RF$), but time ($T_{rest}$) spent resting decreases it. Both of these are scaled for each participant by two corresponding free parameters that define a person's short-term fatigability ($\alpha, \delta$). Unrecoverable fatigue also increases through exerted efforts, but never declines. This is also weighted by an idiosyncratic free parameter ($\theta$), which defines long-term fatigability. These are summed to create $F$, which then serves to increase the discounting of rewards by effort as they increase. Models 3 and 4 include only the effects of $UF$ or $RF$. **b** Schematic representation for how $F$ (both $UF$ and $RF$ combined) affect value and choices to work or rest, with greater discounting of rewards by effort as fatigue increases. **c** Model comparison highlights that the full model is a better predictor of choice data than the simpler models in terms of AIC (left) and in exceedance probabilities (right), highlighting that both $RF$ and $UF$ are necessary to understand the willingness to exert effort. Star indicates winning model. **d** Furthermore, the model-estimated $RF$ and $UF$—from model 5—significantly interacted with effort to predict choice behaviour in a logistic regression. The asterisks show significant $t$-scores (two-tailed $p < 0.001$) and the error bars represent SEM; $n = 36$.

explanatory power. In addition, to test whether the full model was the most frequent in the population we calculated exceedance probabilities for each model[35]. The full model had the highest probability of being the most frequently best fitting model to participants' choice data (Fig. 3 and Supplementary Table 5). This would suggest that participants made decisions to work based on a fatigue-weighted value, where fatigue depended on recoverable and unrecoverable states. Notably the UF parameter ($\theta$) also correlated with the change in ratings of fatigue taken before and

after the main task ($r_s = 0.361$, two-tailed $p = 0.033$, 95% CI = [0.032, 0.620]) suggesting that choice behaviour may have been linked to sensations of fatigue (Supplementary Results).

To test that this model was not only better than the alternatives but also significantly predicted choice behaviour, we performed a logistic regression on work versus rest decisions including $z$-scored reward and $z$-scored interactions of effort and trial-by-trial model-estimated RF and UF as predictors. As in the previous logistic regressions (with a Wilcoxon test across participants),

reward significantly predicted choice ($p < 0.001$), but crucially there were also significant negative interactions of effort and both fatigue components (RF × effort: $Z = -4.98$, $p < 0.001$, 95% CI = [$-4.93$, $-2.98$]; UF × effort: $Z = -5.17$, $p < 0.001$, 95% CI = [$-5.06$, $-3.39$]). This was the case whether using the average estimated fatigue across participants or the model's idiosyncratic estimate of fatigue from each participant (all $ps < 0.001$). Thus, when the levels of fatigue in the model were higher, it was predictive of a greater tendency to rest, in particular when higher effort levels were on offer. Therefore, the willingness to exert effort for reward is not static but fluctuates moment-to-moment. When fatigue states in our model are higher this is related to a greater discounting of reward by effort.

To further examine whether the computational model was able to capture sensations of fatigue, we performed an additional, similar behavioural experiment (Supplementary Methods; Supplementary Fig. 5). In this study, participants ($n = 40$) performed a task with identical effort (0, 30, 39, 48%) and reward levels (6, 8, or 10 credits) to those used in the main task of the fMRI experiment. However, rather than being able to freely choose whether to work or rest on each trial, instead they were required to exert a level of effort (or take a rest) and then rate their level of 'tiredness' (a synonym for fatigue) on each trial. The computational model would predict that fatigue ratings would (i) increase as a function of effort exerted, (ii) would decrease after a trial of rest, (iii) the build-up would be best characterised by both RF and UF factors and (iv) would change independently of reward. In line with the predictions of our model, we found a significant effect of effort on trial-by-trial changes in fatigue ratings, a significant reduction in ratings after a trial of rest, but no significant effect of reward on ratings (Supplementary Fig. 6). To directly test these claims, we fit the three models that aim to capture changes in fatigue, that were fitted to effort-based choices in the fMRI experiment, to the trial-by-trial ratings in the behavioural study (Supplementary Fig. 7 and Supplementary Table 5). The full model, containing separate RF and UF parameters better explained ratings than the other models. These results support the notion that our model is able to capture trial-by-trial changes in fatigue induced by effort, as well as its effects on the value ascribed to exerting effort for reward.

**fMRI results.** We hypothesised that regions of the brain previously linked to effort-based decision-making would be dissociable in terms of signalling different hidden states within the computational model. That is, the BOLD signal in some regions would fluctuate with levels of recoverable and unrecoverable fatigue. To test this notion, we fitted parametric trial-by-trial regressors of the model-estimated UF and RF time-locked to the moment when participants were presented with the work and rest offers. Although this was the moment when people were evaluating the work offer, these regressors carried information only about levels of fatigue—the history of exerted effort—and thus were not correlated with the effort level and reward of the work offer on the current trial. In addition, we examined activity covarying trial-by-trial with the model-estimated, fatigue-weighted, SV of work time-locked to the same event. These parametric regressors were not strongly correlated ($r < 0.4$, see Methods) and thus activity independently covarying with each could be identified.

**Distinct regions signal hidden recoverable and unrecoverable fatigue states during choice.** To test our hypotheses, we first wanted to examine whether distinct regions signalled fatigue states on different timescales. We therefore examined voxels in which activity at the whole-brain level, and within our

hypothesised regions of interest ([ROI—see Methods), significantly covaried with parametric regressors reflecting the trial-by-trial values of RF, UF and SV. Then we tested whether the same voxels significantly covaried with one parametric regressor and did not significantly covary with the others. Such an approach of examining overlap avoids the problems of double-dipping in ROI based analyses. Thus, results we report reflect a response exclusively to RF and UF.

A t-contrast on RF—to extract beta values corresponding to that regressor—revealed a significant negative relationship between the BOLD signal in a cluster extending across the posterior rostral cingulate zone (RCZp: 9, 5, 50; $Z = 3.57$, $p = 0.038$ small-volume FWE correction [svc]; Fig. 4). T-contrasts on UF and SV did not reveal voxels in this region for either contrast ($p > 0.001$ uncorrected). T-contrasts between RF and UF, as well as RF and SV, revealed significant clusters in the RCZp ($p < 0.05$ svc) overlapping with that showing a significant effect in the contrast on RF. Thus, at the time of evaluating and making effort-based decisions, activity in a region extending across the RCZp covaried negatively with a hidden *recoverable state* of fatigue that is increased through effort but recovered through rest.

A t-contrast on UF revealed a significant negative relationship with the BOLD signal in clusters in the left MFG of the DLPFC (Fig. 4 and Supplementary Table 2) as well as in a cluster spanning the anterior rostral cingulate zone (RCZa) and pre-SMA ($-6$, 20, 47; $Z = 3.67$; $p = 0.030$ svc; Fig. 4). T-contrasts on RF and SV did not reveal voxels in the left MFG or RCZa for either contrast ($p > 0.001$ uncorrected). Contrasts between UF and RF, as well as UF and SV, revealed significant clusters in both the MFG and RCZa, albeit at a reduced threshold ($p < 0.05$), overlapping with those showing a significant effect in the contrast of UF above. Thus, a distinct portion of the ACC from that which signalled RF, showed an effect of UF, as did a portion of the MFG. Moreover, in both of these regions activity did not covary with the other components of the model. Thus, at the time of evaluating and choosing whether to exert effort for reward, RCZa and MFG activity negatively covaried with a gradually increasing state of fatigue that was associated with reductions in the willingness to work. In addition, a similar level of specificity was identified in a region of the insula in which activity positively correlated with UF (30, 17, 14; $Z = 4.67$; $p = 0.026$ FWE).

It has been suggested that signals in some regions linked to effort-based decision-making may signal the difficulty of the decision. Although different metrics of decision difficulty and conflict exist, we used the probabilities calculated by the softmax function as a metric of choice difficulty. Notably, activity in none of the regions outlined above covaried with decision difficulty ($p > 0.001$ uncorrected). Moreover, we did not find that these regions specifically signalled variation in trial-by-trial reaction times ($p > 0.001$ uncorrected, Supplementary Table 4).

**A fronto-striatal system integrating value and fatigue.** Next we examined activity that covaried with the fatigue-weighted value of the work offer estimated by the model (Fig. 5 and Supplementary Table 3), using the same approach outlined above to identify activity that scaled only with SV. A t-contrast on SV revealed a significant positive relationship between the BOLD signal in the superior frontal gyrus (SFG) extending into the FP ($-12$, 68, 17; $Z = 4.67$, $p = 0.025$ FWE) as well as in the VS, with the peak voxel within the nucleus accumbens of the Harvard-Oxford atlas (9, 11, $-10$; $Z = 4.31$, $p = 0.001$ svc). T-contrasts of RF and UF did not reveal voxels in this region for either contrast ($p > 0.001$ uncorrected). Contrasts between SV and RF, as well as SV and UF, revealed significant clusters in both FP and VS ($p < 0.05$ svc) overlapping with those showing a significant effect in the contrast

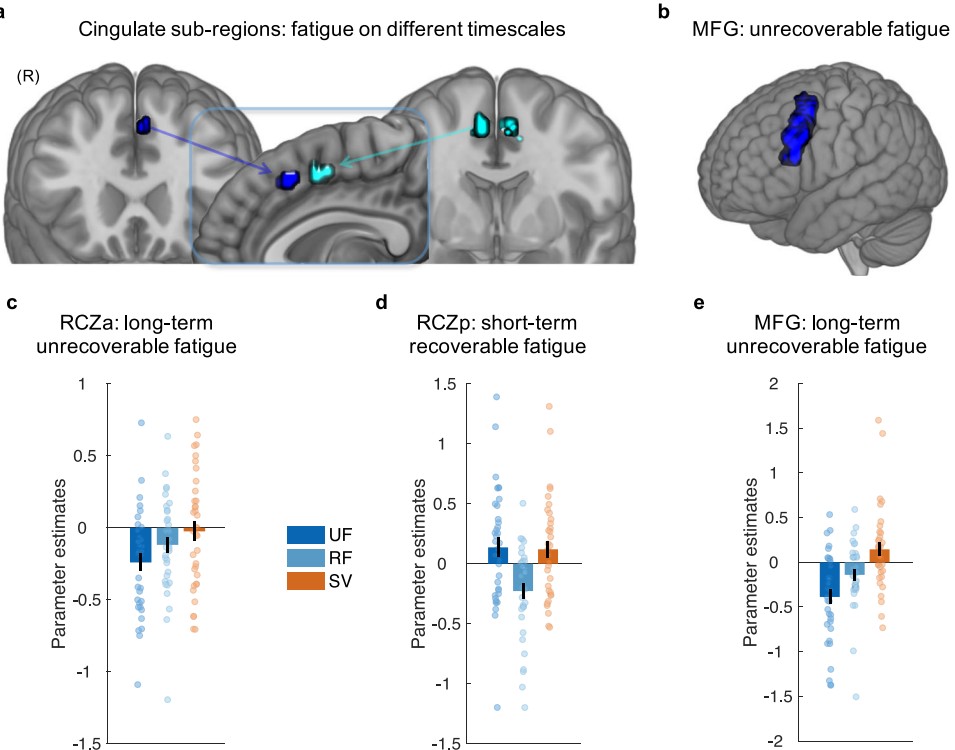

**Fig. 4 Hidden states of fatigue during effort-based decision-making. a** The BOLD signal in two distinct sub-regions of the ACC covaried trial-to-trial with unrecoverable (UF) and recoverable fatigue (RF) states estimated by the model. Overlay of clusters in the anterior Rostral Cingulate Zone (RCZa; dark blue) with activity covarying with UF, and the posterior Rostral Cingulate Zone (RCZp; cyan) with activity covarying with RF. Inset shows non-overlapping clusters. RCZ regions defined with respect to the parcellation of Neubert et al.[43]. **b** Activity in the middle frontal gyri (MFG) covaried with UF. Images displayed at $p < 0.001$ uncorrected. Parameter estimates (arbitrary units) at peak coordinates from RCZa (**c**), RCZp (**d**) and MFG (**e**) for UF, RF and fatigue-weighted subjective value (SV). Each dot represents one subject. Error bars reflect SEM. All $n = 36$.

on SV. Consistent with the idea that activity in the VS was signalling fatigue-weighted value in a subjective manner, we also found significant correlations between the strength of signalling in the VS for UF and RF and each participant's corresponding parameter weights ($\alpha$, $\theta$, $\delta$) from the model. To avoid double-dipping we performed an independent analysis to identify voxels in which SV related activity correlated with parameters from the computational model. We found that the degree to which the VS signalled UF covaried with the UF parameter ($\theta$; 6, 5, −7; $Z = 4.55$; $p = 0.043$ FWE). In addition, the degree to which the VS signalled RF correlated with the effect of rest on the RF ($\delta$; 9, 17, −10; $Z = 3.36$; $p = 0.028$ svc) and the effect of effort on RF, albeit at a reduced threshold ($\alpha$; 9, 17, −10; $Z = 2.93$; $p < 0.005$ uncorrected). No such effects were found in voxels in the other ROIs, with individual differences in fatigue and value only reflected in the VS response. Thus, a distinct fronto-striatal system processed the current value of exerting effort to obtain rewards, integrating momentary levels of fatigue that modulate the value ascribed to working. In the VS, variability between people in the degree to which the fatigue variables covaried with activity, correlated with the parameters from the model that dictated how much someone's willingness to work was under the influence of fatigue.

## Discussion

Many of our daily activities require us to persevere and keep exerting effort to obtain rewards. Here we show that two hidden states, one longer-term unrecoverable and one short-term recoverable, impact on people's decisions to work and exert effort for

reward on a trial-by-trial basis. When these levels of fatigue are higher, it leads to a decrease in the value of working, resulting in choices to rest, particularly when working will be higher in effort and lower in reward. The BOLD response in distinct portions of the frontal cortex covaried separately with these two hidden states, with the MFG and the RCZa signalling the unrecoverable component, and a distinct RCZp region signalling the recoverable component of fatigue. These regions carried no information about the SV of working. Instead, activity in a distinct fronto-striatal system comprising the VS and the FP integrated the latent states to signal the current value of working weighted by levels of fatigue. Moreover, the same computational model captured trial-by-trial ratings of fatigue, such that sensations of fatigue appear to similarly occur on two timescales, suggesting reductions in the willingness to exert effort may directly follow increases in feelings of exhaustion on a moment-to-moment basis. These results highlight that the willingness to exert effort is not static, and changes in fatigue states shift how much value we ascribe to working on a momentary basis. Moreover, different brain regions are involved in dynamically signalling different components of fatigue.

In this study, people's subjective valuations of exerting effort for reward shifted constantly. In particular, choices to work were avoided at reward and effort levels in the main task that participants had readily chosen to undertake in a pre-task where fatigue would not have accumulated. The results suggest that these changes in the willingness to work were reactive, as a result of changes in internal states, rather than related to pre-emptive shifts in valuation. Such a conclusion is supported by the fact that planning was prevented by the experimental design, with offers of

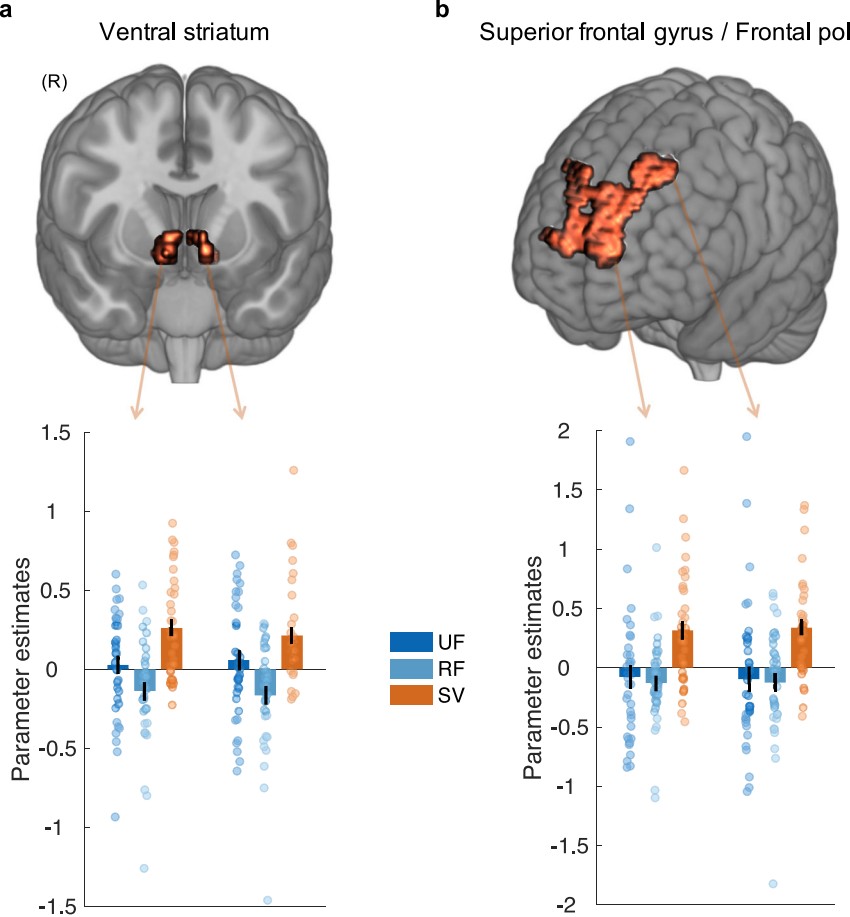

**Fig. 5 Fatigue-weighted subjective value in a fronto-striatal circuit. a** The BOLD signal in the ventral striatum (VS) covaried with model-estimated subjective value (SV), which is weighted by momentary levels of fatigue. Parameter estimates (arbitrary units) from peak coordinates for left and right VS (below) for the responses to SV, recoverable fatigue (RF) and unrecoverable fatigue (UF). **b** BOLD signal in superior frontal gyrus (SFG), extending across SFG/frontal pole areas 9 and 10, signals SV. Parameter estimates (arbitrary units) from peak coordinates in SFG/frontal pole (below) for the responses to SV, RF and UF. Each dot represents one subject. Error bars reflect SEM. Images displayed at $p < 0.001$ uncorrected. All $n = 36$.

effort and reward presented in a pseudorandom order. As such, participants could not plan for the offers to-be-presented in upcoming trials. In addition, our results showed that a model containing fluctuating fatigue components better explained choices than a model in which participants shifted their valuations before the main task, but then held them constant across it. Thus, participants' willingness to work fluctuated and gradually declined across the main task. Offers that participants considered as worth working for at one moment would be rejected in favour of a rest at another.

This study unifies separate lines of research that have theorised that the effects of fatigue may occur on more than one timescale[3,13], and provides a formalised account of their effects on effort-based decisions. One line of research had suggested that extended periods of work lead to exhaustion that has consequences for tasks performed after the one which caused the fatigue[16,25,36–38]. This executive fatigue influences activity in the MFG in tasks performed after having been exhausted, an effect exacerbated in athletes who are over-trained[25,39]. This form of fatigue appears to be unrecoverable in the sense that simply taking short rests does not have a restorative effect. Although in this study we cannot fully rule out the possibility that this effect is simply due to time-on-task or boredom effects, we are able to show that it affects both effort-based decisions and trial-by-trial self-reported fatigue ratings, supporting the notion that it is a

component of fatigue that changes, and not only other subjective effects. Moreover, we show that this longer-term effect is independent from a short-term recoverable component and that it covaries with activity during effort-based choice in the left MFG, largely overlapping with an area which has previously been associated with subjective aversion to cognitive effort[40].

Our results shed new light on this long-term, unrecoverable state. We show that this component is indeed processed in the MFG during effort-based choice but builds slowly during demanding tasks, reducing the willingness to exert effort for reward over an extended time period. Moreover, this effect is localised not only to the MFG but also to a connected sub-region of the cingulate lying in the RCZa[41–43]. Lesions to this region reduce the willingness to exert effort in rodents[44] and neurophysiological recordings here have revealed neurons that respond to effort costs[45]. Taken together, these findings would suggest that the MFG processes a longer-term accumulating fatigue that impacts both effort-based decision-making, performance, and choice behaviour in other tasks[39]. In contrast, the RCZa processes similar information, but perhaps more specifically when deciding whether it is worth exerting effort. Moreover, the fact that activity negatively correlated with UF in RCZa with higher activity when fatigue was lower, and previous evidence that stimulating the RCZa causes a sensation of a willingness to persist through oncoming challenges[46], suggest the RCZa may

play a key role in sustaining motivation and persisting during effortful tasks.

In addition, our results highlight a short-term fatigue effect that crucially is recovered by taking rests. Such a component had long been theorised in accounts of physical fatigue[9,13,47]. However, to date no study had directly examined the changes in neural activity that covary with changes in RF when people are choosing whether a reward and effort are worth it. A previous study had examined how people gave up and returned to work during continuous grip force[9] but did not examine how this influenced effort-based decisions, nor neural activity when ascribing value to work and making an effort-based choice. Here, at the time of making effort-based decisions we found computations of RF in the RCZp. This region has also previously been linked to persistence in decision-making tasks[48–51], but here we show its role in signalling a short-term momentary fatigue state that influences decisions and the value of working.

The findings presented here provide empirical evidence for theories which suggest that the willingness to exert effort fluctuates on a moment-to-moment basis, and they highlight the need for examining such momentary fluctuations to understand variability in cognitive processes over time. The notion of time-on-task effects in cognitively and physically demanding tasks is well known[13,19,24,52–54]. Accuracy and speed decline over time in many effortful cognitive tasks. However, our results suggest that such changes in behaviour may be at least partially driven by a reduction in the value ascribed to persisting with exerting the effort required by the task demands. Considerable research has shown that task performance depends on the balance between the costs and benefits of acts. Rewards can increase the speed and accuracy of both movements and cognitive processes, by paying off the effort costs[48,55]. If the same difficulty of task is treated as more costly over time, as our model predicts, it will devalue rewards to a greater degree, and reduce task performance. Whilst this study evaluated the willingness to work, rather than task performance, the results suggest that there could be moment-to-moment fluctuations in performance due to fluctuations in fatigue happening on multiple timescales. Moreover, they point to a role of the VS for integrating current levels of fatigue with the value of persisting with a demanding task, and variability between people in such tendencies. A limitation of the experiment is that comparing value, without any influence of fatigue, to fatigue-weighted value signals is challenging, as they are necessarily correlated within the design. However, importantly, we found that variability in signalling in the VS between people correlated with the parameters of the computational model. Such a finding is consistent with activity in the VS signalling a dynamically changing estimate of value, which is weighted by each participant's tendency to persist in the face of momentary fatigue. Such effects would be missed or confounded by typical analysis approaches, e.g. when examining changes correlated with trial number or behaviour pre vs post exhaustion, but they can be examined using the formal framework outlined here. Future work will need to understand how the VS integrates fatigue and value-related information leading to fluctuations in the willingness to persist with ongoing behaviour.

A striking aspect of the results was that the different hidden internal variables of fatigue and value mapped on to activity in discrete brain regions, with each area processing information about a distinct component of the model. All of these regions— ACC, MFG, FP and VS—have previously been linked to the processing of effort and reward[4,5,9,10,27,28,30,32,44,51,56–58], but a considerable amount of this work, particularly that focused on the cingulate cortex, had assumed that valuations are static. Our results suggest that dynamic shifts in the valuation of effort correspond to changes in response across two sub-regions of the cingulate cortex, the RCZp and RCZa, and this information is integrated in the VS. Importantly, activity relating to fatigue does not explicitly represent the objective magnitude of the task difficulty. Indeed, responses in these regions covaried with the fatigue variables that carried no information about the reward and effort of the work offer. Rather, when activity in the RCZa and RCZp was reduced, it was at a time when levels of fatigue were higher, *irrespective of the value of the offer*. However, although both regions negatively correlated with levels of fatigue according to the model, they were dissociable, processing fatigue levels on different timescales. Such findings parallel recent evidence from studies examining how different parts of cingulate cortex are activated by learning at different timescales in reinforcement learning tasks[59]. Our results suggest that this may be a wider ranging principal of organisation that extends to other types of decision variables in the cingulate cortex, with different aspects of fatigue shifting how effort and reward are valued in extended tasks.

It was beyond the scope of this investigation to examine whether the different components of fatigue map onto purely psychological changes, physiological or metabolic changes in the state of the body, or fluctuations in neuromodulatory systems[13,17,60]. However, the computational approach taken here was able to best explain changes in decisions about whether to exert effort, and in self-reported sensations of fatigue, and thus are unlikely to be explained by simply time on task or accumulated reward effects. Although accumulated reward and accumulated effort were correlated in the fMRI study, rewards did not influence fatigue ratings trial by trial in the behavioural study. Such findings, that sensations of fatigue were fluctuating in the experiment and could be quantified using the same computational model in which effort exerted causes changes in fatigue, suggest that changes to choice behaviour in the fMRI experiment are more likely to be due to exerted effort than accrued reward. In line with this, the UF and RF components fluctuated in regions that have previously been linked to effort processing, rather than in regions that have been found to signal accumulated reward[61,62]. Future work will need to identify the source of these fluctuating, putative fatigue states, and fully disentangle them from other processes, such as opportunity cost processing, boredom, task switching and time-on-task.

Thus our model may be fruitful for examining such core questions relating to fatigue, including whether similar principles hold when using a cognitively effortful task[8,37,63,64]. By using a model that can quantify, idiosyncratically, each individual's sensitivity to the efforts they have exerted, and to their recovery through rest, variables underlying fatigue can be probed more precisely. Such an approach is also ideally suited for probing fatigue in the multitude of clinical disorders in which fatigue is present[1,65]. Future research may begin to examine how fatigue accumulates and subsides in clinical disorders, in order to develop more appropriate treatments for such a poorly understood symptom of disease.

Overall, this study provides insights into the neural and computational basis of the dynamics of fatigue. The willingness to exert effort fluctuates on a moment-to-moment basis, with shifts in the value of exerting effort for reward depending on a recoverable and an unrecoverable state of fatigue. These states covaried with neural activity in distinct brain regions previously linked to effort-based decision-making, namely in the RCZa and MFG, as well as in the RCZp, when making choices about whether exerting effort is worth for the reward. However, persistence also depended on a fronto-striatal circuit, which integrates fatigue and value, with variability in people's VS response predictive of the influence fatigue had on effort-based choice. These results reveal the hidden determinants of fatigue that underlie persistence in the face of effort.

## Methods

**Participants.** Thirty-nine young, right-handed participants with normal or corrected-to-normal vision and no history of neurological or psychiatric illness were recruited through the Oxford Psychology Research participant recruitment scheme and online bulletin boards. Written informed consent was obtained from all participants prior to the experiment. The study was approved by the University of Oxford Central University Research Ethics Committee (MSD-IDREC-C1-2014-037). One participant did not fully complete the experiment because of feeling uncomfortable in the MRI scanner and was therefore excluded from the analyses, and a further two participants were excluded due to excessive head motion (more than 6 mm of translation). This sample size was selected based on previous fMRI studies in which similar samples evoked responses in hypothesised regions[4,25]. The final sample of 36 participants (16 females) had a mean age of 25.31 years (SD = 4.90; range 18–40). Participants were remunerated with £25 for taking part in the study, plus a possible £10 further as a bonus payment. The bonus depended on the credits accrued on all successfully executed trials of the main task, as well as trials executed during the training and pre-task phase. Thus an increase in the bonus was an incentive on every trial.

**Experimental design.** The aim of this experiment was to examine the hidden states that lead to changes in the willingness to exert effort over time and their neural correlates. We developed a physical effort-based decision-making task, in which effort was operationalised as the amount of grip force that needed to be exerted for a sum total of 3 s in order to obtain reward (credits). The main task of the experiment was performed inside the MRI scanner, but in total the task consisted of four different phases:

i. *Calibration Phase* - To calibrate the levels of effort (grip force), accounting for individual differences in grip strength, each participant's MVC was measured at the beginning of the experiment by squeezing a hand-held dynamometer on three consecutive trials with their dominant (right) hand. Participants were required to apply as much force as possible on each trial, and they received strong verbal encouragement while squeezing. During each attempt, a bar presented on the screen provided feedback of the force being generated. In the second and third attempts, a benchmark representing 105% and 110% respectively, of the previous best attempt was used to encourage participants to improve on their score. The maximum level of force generated throughout the three attempts was used as the participant's MVC to calculate levels of force required for each participant at each effort level.

ii. *Training Phase* – Participants were able to familiarise themselves with the effort levels used across the experiment and learnt how many segments of a pie chart represented each level of force. Participants practiced reaching each of six effort levels (0, 30, 39, 48, 57, and 66% of each participant's MVC). A successful trial occurred only when the force generated by the participant exceeded the required level for a sum total of at least 3 s in a five-second window. Practice of the effort levels was repeated three times, resulting in 18 practice trials in total. Each trial commenced with a pie chart, with the number of red segments indicating the upcoming effort level. During the exertion period, participants were presented with a vertical bar, providing them with real-time feedback on their force and indicating the target effort level by a yellow line superimposed on the bar. When it was a rest trial, indicated by one element in the pie chart, the bar was presented for the same amount of time but with the yellow line displayed at the bottom of the bar. To make sure that participants carefully and successfully completed this training, they were awarded one credit for each successful squeeze, but 0 credits for a failure.

iii. *Pre-task* – Participants performed 75 trials of an effort-based decision-making task before entering the scanner, aimed at measuring participants' devaluation of rewards (credits) by effort in a situation where they would not be experiencing higher levels of fatigue. Participants were required to decide whether they find the rewards on offer are worth the required effort by making a series of choices between two alternatives: a rest (baseline) option for a low reward (1 credit), and an effort (work) option for a higher reward. Work options consisted of one of five different effort levels, represented by two to six filled segments in a pie chart cue that corresponded to 30, 39, 48, 57, and 66% of each participant's MVC, and one of five different reward levels (2, 4, 6, 8, 10 credits) numerically displayed below the pie chart. The rest option was represented by one filled segment in a pie chart and "1 credit" numerically displayed below it. The presentation side (left versus right side of the screen) for 'work' versus rest options was counterbalanced across offers and trials. Effort and reward levels for the 'work' option were varied independently and presented in a pseudorandom order to ensure that each effort/reward combination was distributed evenly across the task. Responses had to be made within 3.5 s, using their left hand on a button box. Otherwise, "0 credits—Make your choices faster" was written on the screen for the time that participants would have spent working or resting. If participants chose to 'work', they were required to exert the chosen force on the dynamometer at least 3 out of 5 s in order to receive the credits associated with the work offer. In this pre-task, only a random 10% of trials actually resulted in the requirement to

work or rest if chosen. Otherwise, a blank screen was presented instead of the work or rest screen and the outcome screen for the equivalent duration. Participants were instructed of this before the beginning of the task, but they were not told which of the choices would count and which ones would not, and they were instructed to always make their decisions as if they would have to squeeze if they chose the work option. Furthermore, this part included two breaks, i.e. was split up into three blocks, and participants were free to decide when to continue with the task. Thus, levels of fatigue would not be induced in the pre-task. Following this, participants received feedback on each trial regarding their success or failure. If all the required force was exerted for 3 s they would receive all of the credits, if they failed to meet this requirement they would receive 0 credits. The choice period in the following trial was separated from the outcome period in the preceding trial and the successive work/rest period by a random jitter of 2–4 s.

iv. *Main task* – This task was performed inside the MRI scanner, aimed at measuring how participants' valuations of effort and reward change across time. Participants performed 216 trials in the same pseudorandom sequence of a task similar to the pre-task, that differed in three ways. Firstly, the range of 'work' offers was lowered to include only those of high value, the three lower effort levels (30, 39, 48% MVC) and the three higher reward levels (6, 8, 10 credits). This ensured that if participants chose to rest, they were doing so for options that they would choose to work for in the pre-task. This would indicate a shift in the value ascribed to working, which was indeed the case in the behavioural data. Secondly, in the main task every choice counted. Thus, every choice to work resulted in the requirement to exert the required level of effort to obtain the offered reward. Thirdly, there were no breaks included. If participants wanted to take a rest, they were instructed they had to choose to do so. The duration of all trials (except for jittering) was the same regardless of choices to rest or work, and across the pre and main task. This ensured that participants' choices and associated neural correlates were not due to temporal discounting[66–68]. Participants were very successful across the whole experiment at reaching the required force levels, successfully obtaining credits on M > 96.8% (SD < 4.6) of the respective trials, suggesting that participants' choices were unlikely confounded by outcome uncertainty. The effort levels used in this experiment were chosen as they have been shown not to cause significant build-up of lactate and muscle pain, and the stopping of exertion is driven more by the perception of effort and not pain[22,69]. This ensured that our results are unlikely to be due to muscle pain, which can be incurred at higher levels of grip force. In addition, prior to the main task and at the end of the experiment participants were required to rate between 0 and 10 their level of 'tiredness'. One participant was excluded from analyses of these ratings as data were not appropriately captured.

**Behavioural analysis.** To determine whether the willingness to exert effort changes as a function of effort, reward, and the history of effortful exertion, logistic regressions (using Matlab's *glmfit* function) on choices, with offered effort and reward levels of the work option as predictor variables were conducted for each participant. To analyse choices in the main task, cumulative effort (the sum total of effort exerted during the task prior to the current trial), as well as interactions of effort and reward levels with cumulative effort (z-scored) were added as additional predictor variables. All regressors were z-scored for each participant, i.e. mean centred and divided by the standard deviation. Regression models were fitted to each participant's choice data, and statistical inference was made at the group level by comparing t-scores across participants against zero. Beta values for each participant's regression coefficients were normalised to t-scores as β/SE(β) in order to compensate for the possibility of poor estimates of βs in participants with low levels of variance. Because the t-scores were not normally distributed, they were tested for significant deviation from zero using two-tailed non-parametric Wilcoxon signed-rank tests. Confidence intervals (CIs) for Wilcoxon tests are based on the Hodges–Lehmann estimate (median).

**Computational modelling**

*Modelling subjective value.* Theoretical accounts and existing empirical data have suggested, but largely not formalised, the notion that fatigue can influence motivation on multiple timescales. Here, we developed a computational model of fatigue based on theoretical accounts and integrated it into a parabolic effort-discounting model to explain how effort-based decisions change over time due to hidden recoverable and unrecoverable components. This model could be fitted separately to each participant's behaviour. In line with previous work on how rewards are parabolically discounted by physical effort[4,33,34], we fitted a simple discounting model to the pre-task choice behaviour. The model assumed that the value of the work offer depends on how rewarding it is, how much effort is required and how participants subjectively weigh these to guide their choices to work or rest. That is:

$$SV_{(t)} = R_{(t)} - (k * E_{(t)}^2) \qquad (1)$$

where $SV_{(t)}$ represents the SV of the work option on trial $t$, and $k$ the subject-specific discount parameter, scaling the devaluation of a reward ($R$, reward level 2,

3, 4, 5, or 6) by the effort ($E$, effort level 2, 3, 4, 5, or 6) required to obtain the reward. The higher an individual's $k$ parameter, the steeper an individual's discount function, i.e. the more this individual's valuation of rewards is discounted by the effort required to obtain the rewards. To fit the model to the data, we used a softmax function, which estimates the probability $P_{(i,t)}$ that a participant will choose the work option $i$ that has a $SV$ over the rest option that has a value of 1 (1 credit, no effort), defined as:

$$P_{(i,t)} = \frac{e^{SV_{(i,t)}*\beta}}{e^{1*\beta} + e^{SV_{(i,t)}*\beta}} \quad (2)$$

Since the baseline $SV$ was fixed at 1 (one credit, no effort), when the baseline was chosen $P_{(i,t)}$ was calculated according to $P_{(i,t)} = 1 - P_{(i,t)}$. Maximum likelihood estimation, using *fminsearch* function in Matlab, was used to minimise the difference between each participant's actual choices and the model estimates for each trial, i.e. to minimise the negative log-likelihood ($LL$). This fitting procedure was used to fit choices in both the pre-task and main task.

The estimates of the discounting parameter $k$ and the level of stochasticity in the choices ($\beta$) were restricted not to go below 0.0276 (in which case even the combinations of lowest reward and highest effort are always accepted) and 0, respectively. The model was fitted 50 times using different random starting values (using *rand*) to ensure that the optimisation function had not settled on a local minimum. By fitting this model to the pre-task, we were able to quantify a participant's typical willingness to exert effort for reward, and the noisiness in such choices, during a task that would not evoke fatigue. The $k$ and $\beta$ parameters obtained for each participant in the pre-task were used as fixed parameters in the models fitted to choices in the main task.

*Modelling fatigue-weighted subjective value (full model).* Based on theoretical accounts we hypothesised that fatigue would increase with exerted effort, would be partially recoverable and decrease with time spent resting, but would also have a gradually increasing unrecoverable component which did not recover with rest[9,13,20,25]. This fatigue impacts value, such that when the levels of fatigue were higher, participants would be less willing to work. Thus, we developed a model including recoverable and unrecoverable components of fatigue that would fluctuate over the experiment and integrated them into the value-based model in Eq. 1:

$$SV_{(t)} = R_{(t)} - ((RF_{(t)} + UF_{(t)})*k*E_{(t)}^2) \quad (3)$$

In this full model, rewards ($R$) are devalued by effort ($E$), subjectively weighted by the discount parameter $k$ from the pre-task. In addition, this discounting effect fluctuates trial-to-trial by levels of recoverable ($RF$) and unrecoverable ($UF$) fatigue. $RF$ subjectively increases if a person exerts effort, i.e. accepts the work offer (Eq. 4), with the work parameter $\alpha$ scaling the amount that effort increases $RF$, and subjectively recovers by time resting ($T_{rest}$), as captured by the rest parameter $\delta$ (Eq. 5). These individual parameters scale how fatigable a person is. $UF$ subjectively accumulates depending on the effort exerted across the whole task, scaled by parameter $\theta$, and is not restored by resting (Eq. 6):

$$RF_{(t)} = RF_{(t-1)} + (\alpha*E_{(t-1)}) \quad (4)$$

$$RF_{(t)} = RF_{(t-1)} - (\delta*T_{rest(t-1)}) \quad (5)$$

$$UF_{(t)} = UF_{(t-1)} + (\theta*E_{(t-1)}) \quad (6)$$

The SV $SV$ and the fatigue levels $RF$ and $UF$ were updated for each trial (initial $RF$ and $UF = 0.5$) and fed into the softmax (Eq. 2) as above, to estimate $P$ in each trial. Based on theoretical considerations, only parameter values $\geq 0$ and $RF$ estimates $\geq$ initial $RF$ were allowed. Missed trials, which were very rare ($M = 0.57\%$ of all trials, SD = 1.71), were treated as rest trials. To maximise the chances of finding global rather than local minima, parameter estimation for the full model and for all alternative models (see below) was repeated over a grid of initialisation values, with 12 initialisations (ranging from 0 to 1.1) per parameter. The optimal set of parameters for each model was used for model comparison and for further analyses.

*Model comparison.* To verify whether the three parameters used to quantify the effects of fatigue were necessary, alternative models were also fitted to participants' choices in the main task. These models fit within a factorial structure of models containing no effect of fatigue (two null models), an effect of $UF$ only (i.e. $\theta$ being fitted), an effect of $RF$ only (i.e. $\alpha$ and $\delta$ being fitted) or the full model with both $RF$ and $UF$. The two null models predicted no effect of fatigue in the main task, one which used the original pre-task discounting parameter ($k$) and thus assumed that the willingness to exert effort stayed the same across the whole experiment, and a second where a new discounting parameter ($\gamma$) was calculated across all trials in the main task which assumed a fixed change in the willingness to work between the pre and main tasks. In addition, two other mathematically plausible, but theoretically unlikely, models were included which used only one parameter to scale the effect of effort and rest on $RF$ (i.e. only $\alpha$ being fitted across both work and rest trials). In one of these models this fatigue was only comprised by this one parameter $RF$, while in a second model, fatigue comprised $UF$ plus the one parameter $RF$. These two models had higher AIC values and thus worse fits than versions of the $RF$ model including separate parameters and are thus not shown in figures. In the

models including a fatigue term, initial $RF$ and $UF$ values were defined such that the initial total fatigue was always equal to 1.

In order to investigate the models' relative ability to predict the behavioural data, model fits were compared using the Akaike Information Criterion (AIC)[70] and Bayesian Information Criterion (BIC)[71] with lower values indicating better fit. Model fit to a given data pattern can be improved by simply adding additional parameters, and thereby models with more parameters may be overfitted. AIC and BIC punish models with more free parameters and favour the most parsimonious solutions by adding a penalty term to the $LL$ which depends on the number of parameters ($d$) and in the case of BIC also on the number of observations, i.e. the number of trials ($n$):

$$AIC = -2*LL + 2*d \quad (7)$$

$$BIC = -2*LL + d*\ln(n) \quad (8)$$

### Functional imaging and analysis

*fMRI scan acquisition.* For anatomical localisation, a high-resolution, three-dimensional structural T1-weighted image was acquired using a magnetisation-prepared rapid gradient echo sequence with 192 slices [slice thickness = 1 mm, repetition time (TR) = 1900 ms, echo time (TE) = 3.97 ms, flip angle = 8°, field of view = 192 mm × 192 mm, voxel size = 1 × 1× 1 mm]. A total of 2355 whole-brain functional T2*-weighted echo planar images (EPIs) were acquired with a tilted-plane sequence with a pitch of 30°, in order to reduce potential image distortions and signal losses caused by susceptibility gradients near air/tissue interfaces[72], using multiband factor acceleration interleaved slice acquisition [72 slices, slice thickness = 2 mm, TR = 1570 ms, TE = 30 ms, flip angle = 70°, field of view = 216 mm × 216 mm, voxel size = 2 × 2 × 2 mm]. Subsequent to the functional sequence, a gradient echo field map sequence was used to collect phase and magnitude maps ($TE_1 = 4.92$ ms, $TE_2 = 7.38$ ms) in order to correct for geometric distortions caused by magnetic field inhomogeneities.

*Image preprocessing.* Imaging data were preprocessed and analysed using Statistical Parametric Mapping (SPM12, Wellcome Department of Imaging Neuroscience, University College London, https://www.fil.ion.ucl.ac.uk/spm). First, to correct for head motion within participants, each EPI in a participant's time-series was realigned to the mean image using a least squares approach and a six parameter, rigid body spatial transformation[73] with B-spline interpolation. In addition, the acquired field maps were used to estimate the amount of non-linear distortion from magnetic field inhomogeneities for each functional image and to correct for the movement-by-distortion interactions[74,75]. Following this, the mean of the realigned and unwarped functional images was coregistered to each participant's own structural image, based on Collignon et al.[76], to ensure better anatomical localisation and greater precision in spatial normalisation. Next, the coregistered structural image was segmented into tissue probability maps based on standard stereotaxic space (Montreal Neurological Institute, MNI), bias-corrected and normalised to the MNI template[77]. The same normalisation parameters were used to convert the realigned and unwarped functional images into standard space. Functional images were then resampled to a voxel size of 3 × 3 × 3 mm and spatially smoothed using an 8 mm full-width-half-maximum Gaussian kernel in order to improve the signal-to-noise ratio.

*First-level statistical analysis.* First level whole-brain statistical analyses were performed using general linear models. To examine whether BOLD activity in any voxel parametrically varied with the trial-by-trial estimates of SV, RF and UF from the winning computational model (full model) during decision-making, the averaged and z-scored trial-by-trial estimates for SV, RF, and UF were used as parametric modulators for the offer cue event-related regressor, i.e. the onset of the options screen. To improve overall model fit and account for potential confounds, the design matrix included the following additional regressors that were not analysed: Four regressors that modelled the onset of the work/rest screen and the onset of the outcome screen, both separately for work trials and rest trials, as well as a regressor that included all events onsets from trials with a missed response.

Regressors were modelled with a stick (delta) function with 0 s duration, convolved with a canonical hemodynamic response function (HRF). For the parametric modulators, a 1st order modulation was selected, i.e. it was assumed that the stick function heights will change linearly over different values of each parametric modulator. Parametric modulators were not orthogonalised with respect to each other. To ensure that the model could be estimated and that respective inferences could be made, the regressors of interest were checked for rank deficiency and statistical independence. Correlations between parametric regressors were all below 0.4 ($r = -0.08$ between RF and UF; $r = -0.09$ between RF and SV; $r = -0.36$ between UF and SV). The six rigid body motion parameters estimated during the realignment step (three translations and three rotations) were added as separate regressors that were not convolved with the HRF to control for nuisance effects resulting from head motion. The high-pass filter cut-off was set to 128 s in order to remove low-frequency noise. Regression coefficients were estimated using a restricted maximum likelihood algorithm, using an autoregressive AR(1) model to account for autocorrelations intrinsic to the fMRI time series. Contrasts for each of the three parametric modulators as well as contrasts between them were conducted to identify regions in which the BOLD signal covaried with each regressor independently and in comparison to each other.

In addition, we ran a control analysis in which we added an index of trial-by-trial decision difficulty as a parametric modulator to the design matrix, time-locked to the onset of the offer cue. Decision difficulty for each participant was calculated as $-|P-0.5|$, with $P$ representing the choice probabilities derived from the softmax function from the full model. That is, more difficult decisions should be reflected as probabilities closer to 0.5 (or difficulty $= 0$), while easier decisions should be reflected as probabilities closer to 0 or 1 (difficulty $< 0$). Trial-by-trial estimates were then averaged across participants and z-scored. Although this approach is limited by averaging across participants, it ensures that variance is scaled similarly across participants. In addition, to examine activity that covaried with reaction times, we also ran a separate GLM in which only individual trial-by-trial reaction times were included as a parametric regressor (Supplementary Table 4).

*Second-level statistical analysis.* In order to be able to make inferences about the sample population, a random effects second-level statistical analysis was conducted. Therefore, the contrast images from the first level were analysed using one-sample *t*-tests at each voxel for each contrast of interest. T-contrasts were then applied to identify areas in which activity varied statistically with the parametric modulators. To correct for multiple comparisons, we used a statistical threshold of $p < 0.05$ with voxel-level family-wise error (FWE) correction across the entire brain volume. Because previous studies have emphasised the importance of the VS and the dACC/pre-SMA region in processing effort-based decisions, and in order to be able to specifically localise activity to anatomically and functionally distinct regions, we also probed these areas using a priori ROI. Therefore, t-contrasts were conducted at the whole-brain level at an uncorrected statistical threshold of $p < 0.001$, and then a FWE small-volume correction was applied using a combined mask taken from appropriate atlases (bilateral VS: from Harvard-Oxford Atlas; bilateral dACC and pre-SMA: areas RCZa, RCZp and pre-SMA defined through resting-state parcellations of the frontal cortex by Neubert et al.[43]. By combining these masks together we provide a more conservative statistical threshold than individual ROI analyses, balancing possible false negatives that can occur with whole-brain correction. Full tables of results are reported at an uncorrected threshold of $p < 0.001$ in Supplementary Tables 1–3. At this reduced threshold clusters in the VS and RCZp lie within a larger cluster and thus do not show in the list of peak results only.

In addition, to examine whether activity was modulated by how strongly a participant's choices were affected by RF and UF, each participant's UF parameter, RF work parameter and RF rest parameter from the full model were used as a covariate for the respective UF and RF t-contrasts in separate second-level group analyses. To avoid double-dipping when correlating parameters with voxels which are already known to show a significant result in a non-independent analysis, we performed these analyses by examining whether any voxels showed a significant effect within the ROI masks. Such an approach does have the limitation that the significance between correlations cannot be determined formally. For these analyses, we excluded one participant who had excessively high RF work and rest parameters.

**Reporting summary**. Further information on research design is available in the Nature Research Reporting Summary linked to this article.

## Data availability

Source data for Figs. 2b, 2f, 3d, 4c-e, 5a-b and Supplementary Figs. 2a, 3, 4c, 6a-c are provided with this paper, and are also available on the Open Science Framework (OSF; https://osf.io/xr84w/; Digital Object Identifier: DOI 10.17605/OSF.IO/XR84W). Unthresholded statistical maps underlying Figs. 4 and 5 are available at the same link. Further fully anonymised behavioural and fMRI data that support the findings of this study are available from the corresponding authors upon reasonable request. Source data are provided with this paper.

## Code availability

Main experimental code is available at this link (https://osf.io/xr84w/; Digital Object Identifier: DOI 10.17605/OSF.IO/XR84W). Custom Matlab code to implement the computational models is available from the corresponding authors upon reasonable request.

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

## Acknowledgements

T.M. was funded by doctoral scholarships from the German Academic Scholarship Foundation (Studienstiftung) and the German Academic Exchange Service (DAAD) as well as by a Funds for Women Graduates (FfWG) Foundation Grant. M.H. was funded by Principal Fellowships from the Wellcome Trust (098282/Z/12/Z, 206330/Z/17/Z). M. A.J.A. was funded by a Biosciences and Biotechnology Research Council (BBSRC) Future Leader Fellowship (BB/M013596/1) and a BBSRC David Phillips Fellowship (BB/ R010668/1). We would like to thank Dr. Campbell Le Heron for his help in programming the fMRI study, Dr. Yuen Siang Ang for assistance with image preprocessing code and Mindaugas Jurgelis for helpful discussions on collecting and modelling fatigue ratings. The Wellcome Centre for Integrative Neuroimaging is supported by core funding from the Wellcome Trust (203139/Z/16/Z).

## Author contributions

T.M., M.A.J.A., and M.H. designed the studies; T.M. collected data; T.M., M.A.J.A., M.C. K.-F., and S.M. contributed to analysis code; T.M., M.A.J.A., and M.C.K.-F. analysed the fMRI study; T.M. analysed the behavioural study; T.M., M.A.J.A., and M.H. wrote the paper.

## Competing interests

The authors declare no competing interests.
