## [Peer Review File · Nature Communications]

REVIEWER COMMENTS

Reviewer #1 (Remarks to the Author):

In this paper the authors use a choice task, computational modeling and fMRI to probe the neural mechanisms of decision making in which agents must explicitly weigh rewards versus effort. Based on previous work, the authors hypothesize that there are two latent processes that influence the tradeoff between rewards and effort: the first is short-term fatigue (RF) that can be diminished by rest; the second a longer-term fatigue (UF) that accumulates over time and that cannot be diminished by rest. They operationalize this hypothesis in a computational model that they fit to subjects' behavior. Further, they use the individual subjects' trial by trial estimates of these processes (as well as the resulting subjective value) to form regressors for use in modelling the fMRI data. They find that the "full" model including both processes fits the subject behavior best (out of a restricted universe of (almost) nested models), that the two processes have significant fMRI BOLD correlates in MFG (RCZa, RCZb: rostral cingulate zone, anterior, posterior), while subjective value has correlates in ventral striatum and superior frontal gyrus. Finally, activity in VS related to RF covaries with individual subjects' RF parameter in the full model, and also with the individual subjects' "rest" parameter.

General Comments

This is an interesting and mostly well-written paper. By nature, the computational modeling approach in which a winning model is selected from a universe of models is basically only as good as the universe of models; this universe is well-motivated theoretically (and the latent processes time series estimates also predict choices independently of reward and effort) lending weight to the conclusions. The neural results for RL and UL are persuasive. The neural results for subjective value, while perhaps not surprising (except for the covariation with behavioral model parameters), bolster the results.

Concerns

1. There is a general sloppiness in language about effort and motivation. Care should be taken to be as precise as possible about these and related terms from the beginning. While the authors operationalize their ideas quite well, there are both hierarchical and contingent ideas presented and I found myself having to turn back quite a bit to make sure I had their idea clear. More effort on this front would help the paper.
2. The winning model doesn't win by "much". How do you explain that the model leaving out UF seems to do "almost as well" as the winning model. It would be helpful to quantify these statements about closeness.
3. Have you considered heterogeneity amongst the subjects with respect to the models? Does the winning model win for the majority of the subjects considered individually?
4. The notation etc. for the models, including especially the time indices, is confusing. I think it would be clearer to define $F_{\{t\}} = RF_{\{t\}} + UF_{\{t\}}$, then give the recursion relations for RF and UF separately, while taking care to be sure that the time indices are defined correctly. Currently, looking at $t = 1$ requires, for example, $UF_{\{-2\}}$, which is not correct. Also, did you consider defining $F_{\{t\}} = 1 + RF_{\{t\}} + UF_{\{t\}}$? Then for $t = 0$, $F_{\{0\}} = 1$, which reduces the (for example) full model to the "bare" no fatigue model, and also means you can make the starting values of the recursion 0 in all cases (for the full model using your definition you state that $UF = RL = 1/2$, but I don't think you state the values in the other cases).

Minor Concerns

1. Pg 2, Line 7. "One increases ..." States don't really increase. Consider different word usage.
2. Pg.2, Line 8. "two states were localized..." The states were not localized.

3. Pg. 3, Line 12. "motivation can reduce" Awkward usage – consider "motivation can decrease" or "motivation can wane".
4. Pg. 4, Line 15. "ventro-medial prefrontal..." . Did you want to define as (VMPFC)?
5. Pg. 4, Line 24. "hypothesise ..." Consider listing the areas here, up-front.
6. Pg. 6, Figure 2 Legend. Panel D) is not explained very well.
7. Pg. 9, Line 19. "logistic regression .." was this on pooled data? If so, why not a random effects model?
8. Pg. 11, Figure 3 Legend. In C) please make it explicit that the model was fitted separately for each subject.
9. Pg. 12, Line 15. Did every subject see the same sequence of choices? If so, could this introduce spurious effects?
10. Pg. 14, Figure 4, Legend. Please define RCZ a,b.
11. Page 15, Line 4. Do you mean to say contrast against baseline? This is activation for a parametric regressor so for each subject the "contrast" taken to the second level should simply be the betas associated with that regressor – no contrast needed.
12. Page 16, Line 19. " ... influence of fatigue." Did this not work in MFG?

Reviewer #2 (Remarks to the Author):

The manuscript describes an fMRI study that investigates the neural foundations of fatigue during a choice task, which involves trading monetary reward against physical effort (handgrip squeeze). The claims are that 1) computational analysis of choice behavior reveals the existence of two fatigue states, one recoverable (RF) and the other unrecoverable (UF), 2) that the neural correlates of these two fatigue states can be dissociated using fMRI and 3) that fatigue modulates the willingness to work encoded in fronto-striatal circuits.

This is an interesting and timely paper. However, although I like the story, I am not convinced that there is enough evidence to make any of these claims (see reasons below). However, the dataset could be exploited to support more reasonable conclusions that would still be worth publishing, in my opinion, although claims of novelty should then be toned down.

Behavior (claim 1):

1) The concept of "moment-to-moment fluctuations in motivational fatigue" is both vague and misleading. What could be at play in the task is simply muscular fatigue: pain in the arm muscles would increase with repeated squeezing and possibly decrease with rest. A participant feeling squeezing as more painful would naturally be less inclined to do it again. The alternative would be that the participant is demotivated, in the sense that the marginal value of additional money would not be worth the effort anymore. Unfortunately, the concept of "motivational fatigue" confounds the two hypotheses, which could be teased apart by comparing models in which what is accumulated is money versus effort, and/or models in which what is impacted is the benefit versus the cost of effort.

2) The evidence in favor of model 5 (with both RF and UF) compared to model 4 (RF only) is weak (slight difference in AIC or BIC). Besides, it seems that, as they are written in Fig. 3, model 5 and model 4 are no different. They can both be rewritten as $F(t) = \text{param1} \times \text{sum}(E) + \text{param2} \times \text{sum}(T)$, i.e. some weighted sum of cumulative effort and cumulative resting time. I might have

missed something here, but at the very least this needs clarification.

3) Simpler accounts must be included in the model comparison. The change in the ranges of reward and effort levels, between pre-task and main task, might by itself induce a change in the subjective value function. One possibility is for instance regression to the mean: in both cases participants tend to accept rewards above the mean, or efforts below the mean, which could be mistaken as a fatigue effect. I am aware that choices denote a shift of preference within the main task, but the change in the range might still affect conclusions, because it is not accounted for in model comparison. To fix the issue, an intercept parameter, capturing the difference between tasks, could be introduced in fatigue models.

4) Besides, instead of cumulative effort, or cumulative reward (as suggested above), a simpler function of time-on task, like trial number, should be tested. It could simply be that participants are more and more bored with the task, or willing to go home. A way to show that fatigue is really about effort cost would be to sum quadratic (not linear) effort levels. Also, instead of an effect on effort cost (or reward benefit, as suggested above), trial index or fatigue could impact an additive parameter, which would suggest that they are just less willing to squeeze anymore, irrespective of reward and effort levels.

Neuroimaging (claims 2 and 3)

5) The finding that left MFG decreases with fatigue (be it RF or UF) is convincing, as the cluster shows up in a whole-brain analysis, surviving correction for multiple comparisons. However, its contribution to the shift in preference could be specified. What the analysis shows is that its activity is decreasing with time on task (or fatigue), but the link to choices is not established. Could it be that left MFG is simply less active when effort is declined, which becomes more and more frequent across task trials? Would this be related to shorter deliberation time?

6) On the contrary, the dissociation between RF and UF relates to cingulate zones that do not appear in activation tables, even at uncorrected threshold. They only survive small-volume correction within pre-defined regions of interest that seem quite arbitrary (why not other regions, like the anterior insula?). I think this level of evidence is way too weak to maintain a conclusion such as neural correlates of RF and UF can be dissociated.

7) There is a double-dipping issue when selecting clusters based on regression against RF or UF and then comparing regression estimates extracted from the peak of these clusters. The issue is that the selection is not independent from the comparison, meaning that it is biased towards voxels in which the noise will favor a significant comparison.

8) Showing that activity in neural regions like the ventral striatum correlates with fatigue-weighted subjective value is no proof that fatigue does affect value signals in these regions. This is because subjective value integrates factors (reward and effort levels) that are sufficient to explain the correlation. In other words, VS activity might correlate with fatigue-weighted SV just because it responds to rewards. To prove their point, the authors need to show that neural activity in VS or other regions is better explained by fatigue-weighted SV than by regular SV (without fatigue).

9) There is the same difference between frontal pole and VS as between MFG and cingulate zones: the former activation is convincing because it survives whole-brain corrected threshold, while the latter rely on a priori ROI. However, I would question the 'frontal pole' label, which usually refers to BA 10. From the map on Fig. 5 it seems that the cluster is more dorsal and posterior, more like superior frontal gyrus (sometimes called dorsomedial prefrontal cortex).

Minor issues:

- Introduction and discussion could more focused, at present there are many redundancies, and the links with cited papers are often loose. Also, the novelty of the computational framework is clearly oversold: increasing effort cost with cumulative effort is quite a standard solution.

- Fig. 2D is not particularly useful, as I cannot see the fatigue effect (I presume the plot is meant

to show darker choice probability with progress in the task).

- The analysis in Fig. 3C is at odds with the winning computational model, because RF and UF are now included as additive (significant) regressors, instead of interacting with effort cost. If the idea is to provide further evidence in favor of the model, this is not helpful. It rather suggests that choices are more and more biased towards rest with increasing trial number.

- Significance levels could be added above regression estimates on figures, so readers can easily identify significant factors.

- I did not find any information about how participants were remunerated. This is important to discard the possibility that they simply trade their payoff against time on task, instead of squeezes.

- There are a few typos that need correction (e.g., Fig. 3B: "Schematic representation for how F ... effect value and choices to work or rest").

Reviewer #3 (Remarks to the Author):

In this interesting manuscript the authors explore the role for trial-wise fatigue in the neural computations of effort-based decision-making. The authors present evidence for two distinct fatigue signals that are distributed across various nodes within a fronto-striatal network. Fatigue is an under-studied and poorly understood construct, and this work therefore has the potential to make a significant and innovative contribution. The paper is superbly written and the analytical methods are sophisticated and appropriate. Despite these strengths, I do have some (mostly minor) concerns with aspects of the analysis and some more significant concerns related to interpretation. I have the following comments for the authors to consider:

The most significant problem as I see it is that "fatigue" is an under-specified construct both conceptually and operationally. Specifically, it seems likely that there are likely moderate to high correlations between the parameters representing recoverable fatigue (RF) and unrecoverable fatigue (UF) and other decision-variables. As I understand the task design and computational model, the UF parameter scales the cumulative expenditure of effort. However, it would seem that cumulative expenditure of effort would also be highly correlated with cumulative rewards, as the effortful option always yields greater rewards. Therefore, this parameter could capture diminishing marginal utility of accumulating points over the course of the task. It would also necessarily correlate at least moderately strongly with the mere passage of time. As such, the strict interpretation as a measure of "unrecoverable fatigue" seems hard to justify. One could just as easily think of it as a global "opportunity cost signal" reflecting the additive and/or interactive effects of fatigue, diminished interest in additional points, a desire to finish up the study and move on to other activities, etc., etc., Indeed, such global opportunity cost signals have been predicted in the context of effort (e.g., Kurzban et al., 2014).

Similarly, for the interpretation of the RF parameter as representing "recoverable fatigue", other interpretations seem equally plausible. It would seem this value might also correspond with foraging values, task switching, etc., all of which could be consistent with the observed results in terms of both the computational model and the imaging results in the RCZ. The authors acknowledge this on the one-hand, but still claim that their work shows a unique RF contribution. But without ruling out the possibility that RF is merely tracking with other decision-variables, claiming a unique RF component seems to be an over-reach.

Related to the above, it was a bit surprising not to see subjective report of fatigue and its association with model parameters. While self-reported fatigue has its own measurement limitations, it would nevertheless provide some additional evidence that the putative fatigue parameters are tracking with the subjective experience of fatigue. If these data were collected as

part of this study then they should be included. If not, it could potentially be included in a follow-up behavioral study in a separate sample.

Another potential concern is floor/ceiling effects. It's unclear how much intra-individual variability there was in choices, which could impact interpretability of fatigue-brain relationships. Based on figure 2D, it appears that the effortful option was chosen a very high percentage of the time. At the individual level, if someone chose almost all effortful options, then we might infer that they simply did not find the task very fatiguing, in which case it becomes less clear how to interpret an association between the RF or UF regressor and neural activity. This could significantly influence power if the effective sample size (subjects contributing meaningful variability in choice behavior) is much lower than the actual sample size.

I appreciate the authors' incorporating a control analysis of choice-difficulty analysis. The method used for estimating choice difficulty is sound, but is susceptible to limitations for participants with highly stable choice preferences (one may agonize over a decision while still arriving to a choice consistent with model predictions). This can lead to a dramatically different scaling of trial-wise difficulty values across subjects. The authors appear to have addressed this issue by averaging difficulty values across participants, but I'm not sure this makes sense. Neural activity for the "average" choice difficulty for a particular trial is not necessarily reflective of individual differences. This may partly explain the null effects for this analysis.

It was unclear if proper control comparisons were performed for imaging results. For example, in the two RCZ regions associated with UF and RF, it would be useful to include the additional direct comparisons to confirm a double-dissociation. It could easily be the case that the area of RCZ showing association with RF is only slightly below SVC threshold for UF, and/or vice-versa, which would significantly change the interpretation of sub-regional specificity.

In their justification for the UF/RF distinction, the authors note the work of Blain and colleagues, showing that greater fatigue led to more impulsivity/inconsistency in choice behavior. It might be interesting to test a similar idea in the current data, e.g., by examining comparing choice behavior in early and late trials in the current task.

Minor comments:

Please show discounting curves as well as parameter value distributions.

It would be worth noting that the region of left MFG associated with UF appears similar to the region identified in a task focused on detecting effort aversion (McGuire et al., PNAS, 2010). That might be worth discussing in terms of the interpretation of UF.

It would be worth examining different striatal ROIs, including those associated with motor function. The authors may want to consider using the Choi 2011 parcellation seeds or some other functional parcellation of the striatum to interrogate its role more thoroughly.

For correlations between neural activity and model parameters, please perform comparisons of correlation coefficients (e.g., Steiger test or equivalent) to confirm differences.

It is unclear why tables report uncorrected whole-brain values? It seems based on the text that these values were derived from a whole-brain corrected map.

Reviewer 1

This is an interesting and mostly well-written paper. By nature, the computational modeling approach in which a winning model is selected from a universe of models is basically only as good as the universe of models; this universe is well-motivated theoretically (and the latent processes time series estimates also predict choices independently of reward and effort) lending weight to the conclusions. The neural results for RL and UL are persuasive. The neural results for subjective value, while perhaps not surprising (except for the covariation with behavioral model parameters), bolster the results.

Response:

We thank the reviewer for their positive evaluation of the work. Their comments have allowed us to improve the manuscript and address their concerns.

1. There is a general sloppiness in language about effort and motivation. Care should be taken to be as precise as possible about these and related terms from the beginning. While the authors operationalize their ideas quite well, there are both heirarchical and contingent ideas presented and I found myself having to turn back quite a bit to make sure I had their idea clear. More effort on this front would help the paper.

Response:

We apologise to the reviewer for lack of clarity in terminology. I am sure the reviewer appreciates that definitions of effort, incentivisation, motivation and fatigue differ considerably across and within different sub-fields. In the revised manuscript we have modified the text in multiple places to ensure greater clarity and ensure consistency in terminology, especially in the introduction.

2. The winning model doesn't win by "much". How do you explain that the model leaving out UF seems to do "almost as well" as the winning model. It would be helpful to quantify these statements about closeness.

3. Have you considered heterogeneity amongst the subjects with respect to the models? Does the winning model win for the majority of the subjects considered individually?

Response:

We thank the reviewer for raising these points about the extent to which this model successfully captures behaviour above alternatives – we address points 2 and 3 together here as they are related. Firstly, from a conceptual point we note that the models all try and capture a very similar phenomena – that is five of the models try and capture an increase in fatigue after effort and reduction after rest, and all of the models assume that rewards act

as an incentive, increasing value and the amount of force required serves as a cost decreasing it. It is therefore expected that the winning model would only show a small improvement over others. However, we agree with the reviewer that it is important for our aims that we show that our model comparison is robust, that UF and RF are needed to explain the effects of fatigue and that our full model is the best characterisation of people's behaviour. To address the reviewer's concerns we have now (i) calculated exceedance probabilities (EP) on the AIC values for the model comparison. EPs take a random-effects approach to calculate the probability that a winning model is the most frequent in the population. We find that the winning model has an EP of 0.72, and has a much higher probability than the next best model without the UF component which has an EP of 0.16. These EPs are now plotted in supplementary figure 4. (ii) We have included an additional behavioural dataset to address this concern and those of other reviewers. In this additional data, participants performed a very similar experimental setup, except on each trial they were forced to either work (at the three levels of force in the original experiment) or rest rather than being able to choose to do so. Rather than make choices, on each trial they then rated how "tired" they were between zero and 100. As can be seen in Supplementary figure 5,6 and 7 participants ratings of fatigue increased across the experiment and depended on the effort that had been exerted on each trial. To examine whether our model could explain changes in fatigue ratings as well as choice behaviour, we fitted the five models used in the original manuscript to explain changes trial to trial ratings in the new dataset. Replicating the original modelling results, using Bayesian model comparison we show that the full model containing independent UF and RF components is the best explanation of fatigue ratings in this new data. Thus, robustly demonstrating that fatigue may well have distinct UF and RF components that operate on distinct timelines.

In the revised manuscript we include this additional behavioural experiment and the analyses of it in the supplementary materials, the corresponding supplementary figures 5-7 as well as including the exceedance probabilities from the original dataset in Supplementary Figure 4B:

Main Text (page 11)

To test whether the full model was the most frequent in the population we calculated exceedance probabilities for each model. The full model had the highest probability of being the most frequently best fitting model to participants' choice data (Supp. Fig.4)

Supplementary Figure 4. Supplementary results from fMRI study behaviour. (A) Histogram of proportion of trials on which participants made choices to “rest” out of 210 trials. There is considerable variability in choices both within and between many participants. (B) Exceedance probabilities for the models fitted to the choice data. Exceedance probabilities for models fitted to fMRI behavioural data. Y-axis reflects the probability of being the most frequently observed model in the population. The full model (5) is the most frequently best fitting model in the population. (C) Model parameters for each participant (green dot) and average across participants for the discounting parameter fitted to the pre-task (left), the two recoverable parameters (middle) and the unrecoverable parameter (right). Error bars represent SEM.

Supplementary Figure 5. Behavioural study trial structure. Participants were required to exert force for rewards, with effort levels calibrated to their MVC (A), after training at each of four effort levels (0%,30,39,48% MVC) they performed 120 forced execution trails. On each trail they were instructed an effort that would be required (indicated by a pie chart), and then required to exert that level of force for a total of 3 out of 5s to obtain credits. They were then told the amount of credits received – 6, 8 or 10 credits if successful or 0 credits if failing to exert the required force. Following this they rated their level of tiredness from 0-100 on a continuous scale.

C) Significant effect of effort but not reward on fatigue ratings

Supplementary Figure 6. Behavioural study rating results. (A) Mean trial by trial ratings of “tiredness” between 0-100 across trials. Shaded areas represents SEM. (B) Change in fatigue ratings from t to $t-1$ as a function of effort level (x-axis) and reward (shade of blue) on trial t (i.e. the effort just exerted and the reward received for it). Only successful trials are included. Rest trials (0% MVC) induced recovery, with a linear effect of effort on increasing ratings. (C) Linear regression on trial to trial changes in ratings revealed significant effects of effort, but no significant effect of reward or effort x reward interaction. Error bars depict SEM. Results support the notion of a gradual increase in fatigue (A) across the experiment as well as trial by trial short-term recoverable changes (B).

A) Model comparison reveals best fit by full model

A) Full model is most frequently observed in the population

C) Models tested

Model 1: UF only (no recovery)

$$F_{(t)} = UF_{(t-1)} + \theta * E_{(t)}$$

Model 2: RF only, one parameter

$$F_{(t)} = RF_{(t-1)} + \alpha * E_{(t)} - \alpha * T_{(t)}$$

Model 3: RF, one parameter, and UF

$$F_{(t)} = (RF_{(t-1)} + \alpha * E_{(t)} - \alpha * T_{(t)}) + (UF_{(t-1)} + \theta * E_{(t)})$$

Model 4: RF only, two parameters

$$F_{(t)} = RF_{(t-1)} + \alpha * E_{(t)} - \delta * T_{(t)}$$

Model 5: Full model

$$F_{(t)} = (RF_{(t-1)} + \alpha * E_{(t)} - \delta * T_{(t)}) + (UF_{(t-1)} + \theta * E_{(t)})$$

Supplementary Figure 7. Computational modelling results for behavioural rating study. (A) Model comparison results in AIC for the models predicting changes in fatigue ratings across the experiment. The full model (5) is the best fit to the data when punishing for the number of parameters. (B) Exceedance probabilities for the models fitted to the ratings data Y-axis reflects the probability of being the most frequently observed model in the population. Model 5 is the winning “full” model of fatigue containing separate RF and UF components. (C) Models compared. All models predicted changes in fatigue ratings. The best fitting model predicted changes in fatigue that were partially recoverable - increase through effort, decrease through rest - but also contained a long-term unrecoverable component.

4. The notation etc. for the models, including especially the time indices, is confusing. I think it would be clearer to define $F_{\{t\}} = RF_{\{t\}} + UF_{\{t\}}$, then give the recursion relations for RF and UF separately, while taking care to be sure that the time indices are defined correctly. Currently, looking at $t = 1$ requires, for example, $UF_{\{-2\}}$, which is not correct. Also, did you consider defining $F_{\{t\}} = 1 + RF_{\{t\}} + UF_{\{t\}}$? Then for $t = 0$, $F_{\{0\}} = 1$, which reduces the (for example) full model to the “bare” no fatigue model, and also means you can make the starting values of the recursion 0 in all cases (for the full model using your definition you state that $UF = RL = \frac{1}{2}$, but I don’t think you state the values in the other cases).

Response: We apologise to the reviewer for confusion caused by the notation of the models. We have now considerably revised how the models are described and their notation to ensure consistency and clarity. The reason for the model being implemented in the manner we have, rather than in the manner suggested, is that we wanted the model to also be able to be fitted in the future to other forms of data (e.g. ratings of fatigue, such as those now included the new data outlined in our previous response). Thus, keeping the “1+” out of the model itself allowed it to be fitted with only a starting value – which was fixed at $F=1$ in the decision-making model, but set to an initial rating by the participant in the model fitted to the ratings data. Thus, the models were comparable across the ratings data and the choice data. In the revised manuscript we have also clarified these points and the starting value for all models to ensure transparency. We thank the reviewer for helping us make these points clear.

Main Text (page 28)

Computational modelling

Modelling subjective value

Theoretical accounts and existing empirical data have suggested, but largely not formalised the notion that fatigue can influence motivation on multiple timeframes. Here, we developed a computational model of fatigue based on theoretical accounts and integrated it into a parabolic ‘effort-discounting’ model to explain how effort-based decisions change over-time due to hidden recoverable and unrecoverable components. This model could be fitted separately to each participant’s behaviour. In line with previous work on how rewards are parabolically discounted by physical effort^{5,33,34}, we fitted a simple discounting model to the pre-task choice behaviour. The model assumed that the value of the work offer depends on how rewarding it is, how much effort is required and how participants subjectively weigh these to guide their choices to work or rest. That is:

$$SV_{(t)} = R_{(t)} - (k * E_{(t)}^2) \quad (1)$$

where $SV_{(t)}$ represents the subjective value of the work option on trial t , and k the subject-specific ‘discount parameter’, scaling the devaluation of a reward (R , reward level 2, 3, 4, 5, or 6) by the effort (E , effort level 2, 3, 4, 5, or 6) required to obtain the reward. The higher an individual’s k parameter, the steeper an individual’s discount function, i.e. the more this individual’s valuation of rewards is discounted by the effort required to obtain

the rewards. To fit the model to the data, we used a softmax function, which estimates the probability $P_{(i,t)}$ that a participant will choose the work option i that has a subjective value SV over the rest option that has a value of 1 (1 credit, no effort) defined as:

$$P_{(i,t)} = \frac{e^{SV_{(i,t)} * \beta}}{e^{1 * \beta} + e^{SV_{(i,t)} * \beta}} \quad (2)$$

Since the baseline SV was fixed at 1 (one credit, no effort), when the baseline was chosen $P_{(i,t)}$ was calculated according to $P_{(i,t)} = 1 - P_{(i,t)}$. Maximum likelihood estimation, using *fminsearch* function in Matlab, was used to minimise the difference between each participant's actual choices and the model estimates for each trial, i.e. to minimise the negative log-likelihood. This fitting procedure was used to fit choices in both the pre-task and main task.

The estimates of the discounting parameter k and the level of stochasticity in the choices (β) were restricted not to go below 0.0276 (in which case even the combinations of lowest reward and highest effort are always accepted) and 0, respectively. The model was fitted 50 times using different random starting values (using *rand*) to ensure that the optimisation function had not settled on a local minimum. By fitting this model to the pre-task, we were able to quantify a participant's typical willingness to exert effort for reward, and the noisiness in such choices, during a task that would not evoke fatigue. The k and β parameters obtained for each participant in the pre-task were used as fixed parameters in the models fitted to choices in the main task.

Modelling fatigue-weighted subjective value (full model)

Based on theoretical accounts we hypothesised that fatigue would increase with exerted effort, would be partially recoverable and decrease with time spent resting, but would also have a gradually increasing unrecoverable component which did not recover with rest^{10,11,14,21}. This fatigue impacts value, such that when the levels of fatigue were higher, participants would be less willing to work. Thus, we developed a model including recoverable and unrecoverable components of fatigue that would fluctuate over the experiment and integrated them into the value-based model in Equation 1:

$$SV_{(t)} = R_{(t)} - ((RF_{(t)} + UF_{(t)}) * k * E_{(t)}^2) \quad (3)$$

In this full model, rewards (R) are devalued by effort (E), subjectively weighted by the discount parameter k from the pre-task. In addition, this discounting effect fluctuates trial-to-trial by levels of recoverable (RF) and unrecoverable (UF) fatigue. RF subjectively increases if a person exerts effort, i.e. accepts the work offer (Equation 4), with the work parameter α scaling the amount that effort increases RF , and subjectively recovers by time resting (T), as captured by the rest parameter δ (Equation 5). UF subjectively accumulates depending on the effort exerted across the whole task, scaled by parameter θ , and is not restored by resting (Equation 6):

$$RF_{(t)} = RF_{(t-1)} + (\alpha * E_{(t-1)}) \quad (4)$$

$$RF_{(t)} = RF_{(t-1)} - (\delta * T_{(t-1)}) \quad (5)$$

$$UF_{(t)} = UF_{(t-1)} + (\theta * E_{(t-1)}) \quad (6)$$

The subjective value SV and the fatigue levels RF and UF were updated for each trial (initial RF and $UF = 0.5$) and fed into the softmax (Equation 2) as above, to estimate P in each trial. Based on theoretical considerations, only parameter values ≥ 0 and RF estimates \geq initial RF were allowed. Missed trials, which were very rare ($M = 0.57\%$ of all trials, $SD = 1.71$), were treated as rest trials. To maximise the chances of finding global rather than local minima, parameter estimation for the full model and for all alternative models (see below) was repeated over a grid of initialisation values, with 12 initialisations (ranging from 0 to 1.1) per parameter. The optimal set of parameters for each model was used for model comparison and for further analyses.

Model comparison

To verify whether the three parameters used to quantify the effects of fatigue were necessary, six other models were also fitted to participants' choices in the main task. The alternative models either assumed (i) a UF component only (i.e. θ being fitted), (ii) an RF component only including one parameter, assuming that the rates of fatigue build-up during effort and of fatigue decrease during rest are similar, (iii) an RF component including one parameter as well as a UF component, (iv) an RF component including separate work and rest parameters (i.e. α and δ being fitted) but no UF , and (v) a change in motivation which is not due to fatigue, i.e. a new discount parameter γ was fitted and each individual's k parameter from the pre-task was disregarded. In the models including a fatigue term, initial RF and UF were defined such that the initial total fatigue always equalled 1.

In order to investigate the models' relative ability to predict the behavioural data, model fits were compared using the Akaike Information Criterion (AIC)⁷⁰ and Bayesian Information Criterion (BIC)⁷¹ with lower values indicating better fit. Model fit to a given data pattern can be improved by simply adding additional parameters, and thereby models with more parameters may be overfitted. AIC and BIC punish models with more free parameters and favour the most parsimonious solutions by adding a penalty term to the log-likelihood (LL) which depends on the number of parameters (d) and in the case of BIC also on the number of observations, i.e. the number of trials (n):

$$AIC = -2 * LL + 2 * d \quad (7)$$

$$BIC = -2 * LL + d * \ln(n) \quad (8)$$

Minor Concerns

1. Pg 2, Line 7. *“One increases ...” States don’t really increase. Consider different word usage.*
2. Pg.2, Line 8. *“two states were localized...” The states were not localized.*
3. Pg. 3, Line 12. *“motivation can reduce” Awkward usage – consider “motivation can decrease” or “motivation can wane”.*
4. Pg. 4, Line 15. *“ventro-medial prefrontal...” . Did you want to define as (VMPFC)?*
5. Pg. 4, Line 24. *“hypothesise ...” Consider listing the areas here, up-front.*
6. Pg. 6, Figure 2 Legend. *Panel D) is not explained very well.*

Response: We thank the reviewer for noting these less than ideal choices of words, we have now edited each appropriately in the revised manuscript.

7. Pg. 9, Line 19. *“logistic regression ..” was this on pooled data? If so, why not a random effects model?*

Response: A logistic regression was performed on each participant’s choice data separately, beta estimates were carried through to a second-level statistical inference. The data was not pooled across participants. This has now been clarified in the revised manuscript.

Page 28

Regression models were fitted to each participant’s choice data, and statistical inference was made at the group level by comparing t-scores across participants against zero. Beta values for each participant’s regression coefficients were normalised to t-statistics as $\beta/SE(\beta)$ in order to compensate for the possibility of poor estimates of β s in participants with low levels of variance. Because the t-scores were not normally distributed, they were tested for significant deviation from zero using non-parametric Wilcoxon signed-rank tests.

8. Pg. 11, Figure 3 Legend. In C) please make it explicit that the model was fitted separately for each subject.

Response: This has now been clarified in the figure legend as well as in the main text.

Figure 3

This full model was fitted to each participant's choice data separately and then compared with the fit of the other models, with a model comparison revealing the full model to be the best fit to the data.

Page 28

This model could be fitted separately to each participants' behaviour.

9. Pg. 12, Line 15. *Did every subject see the same sequence of choices? If so, could this introduce spurious effects?*

Response: The reviewer is correct that only one sequence of trials was used, this was because we wanted to examine individual differences between participants in the fluctuations of fatigue in behavioural and fMRI analyses. Multiple sequences may have caused participants with higher parameters to not necessarily be those with greater influences, but occurred due to the sequence of trials they received. However, it is unlikely that a single sequence could explain our results, as Figure 2 *panel d* highlights, there is a strikingly high variability in the trials in which participants chose to rest, and in the model parameter weights shown in supplementary Figure 4. Such variability is likely to reflect a different build-up of fatigue across participants, rather than being a result of the sequence of trials. Moreover, the additional experiment now added to the manuscript used a different trial structure in a different pool of participants but found corresponding statistical and modelling results.

10. Pg. 14, Figure 4, Legend. *Please define RCZ a,b.*

Response: We have now defined the RCZ regions more clearly.

Figure 4 Legend

A) The BOLD signal in two distinct sub-regions of the ACC covaried trial-to-trial with unrecoverable (UF) and recoverable fatigue (RF) states estimated by the model. Overlay of clusters in the anterior Rostral Cingulate Zone (RCZa; dark blue) with activity covarying with UF, and the posterior Rostral Cingulate Zone (RCZp; cyan) with activity covarying with RF. Inset shows non-overlapping clusters. RCZ regions defined with respect to the parcellation of Neubert et al.

11. Page 15, Line 4. *Do you mean to say contrast against baseline? This is activation for a parametric regressor so for each subject the "contrast" taken to the second level should simply be the betas associated with that regressor – no contrast needed.*

Response: we apologise for any confusion. As is standard in SPM, to extract betas for a parametric regressor you run a “contrast” of “1” for that regressor. We have clarified the wording in the revised manuscript.

Page 14

A t-contrast on RF - to extract beta values corresponding to that regressor...

12. Page 16, Line 19. “ ... influence of fatigue.” Did this not work in MFG?

Response: Thank you for raising this interesting point. We did not find such correlations in any region other than the VS, possibly suggesting the VS has a more important role in integrating fatigue to influence subjective value. We have now clarified this in the revised manuscript

Page 16

No such effects were found in any other region, with individual differences in fatigue and value only reflected in the VS response.

Reviewer #2 (Remarks to the Author):

The manuscript describes an fMRI study that investigates the neural foundations of fatigue during a choice task, which involves trading monetary reward against physical effort (handgrip squeeze). The claims are that 1) computational analysis of choice behavior reveals the existence of two fatigue states, one recoverable (RF) and the other unrecoverable (UF), 2) that the neural correlates of these two fatigue states can be dissociated using fMRI and 3) that fatigue modulates the willingness to work encoded in fronto-striatal circuits.

This is an interesting and timely paper. However, although I like the story, I am not convinced that there is enough evidence to make any of these claims (see reasons below). However, the dataset could be exploited to support more reasonable conclusions that would still be worth publishing, in my opinion, although claims of novelty should then be toned down.

Response: We thank the reviewer for saying that the work is interesting and timely. We appreciate their helpful critique of our claims, we have used their comments to refine definitions of concepts, included additional analyses in line with their suggestion and an additional behavioural experiment which more directly shows that people’s sensations of fatigue are reflected in recoverable and unrecoverable components.

Behavior (claim 1):

1) The concept of “moment-to-moment fluctuations in motivational fatigue” is both vague and misleading. What could be at play in the task is simply muscular fatigue: pain in the arm muscles would increase with repeated squeezing and possibly decrease with rest. A participant feeling squeezing as more painful would naturally be less inclined to do it again. The alternative would be that the participant is demotivated, in the sense that the marginal value of additional money would not be worth the effort anymore. Unfortunately, the concept of “motivational fatigue” confounds the two hypotheses, which could be teased apart by comparing models in which what is accumulated is money versus effort, and/or models in which what is impacted is the benefit versus the cost of effort.

Response: We apologise to the reviewer that our claims regarding fatigue were not fully clear. There are a number of points to unpack in order to comprehensively reply, so we have broken our response down to address each of these points separately:

- a) To fully clarify our definition – which also underpins our modelling approach – our claim is that the motivation to exert effort involves rewards being devalued by effort. However, theories suggest that sensations of fatigue putatively modulate this, such that when people are more exhausted they show an increased devaluation effect. We refer to these fluctuations in motivation driven by momentary changes in fatigue, as motivational fatigue in order to be concise. Although included in theoretical accounts, these ideas had not been tested in experiments that examine motivation on a trial-by-trial basis, with a model that predicted changes in effort discounting from one trial to the next (i.e. examined momentary changes in motivation that might be under the influence of fatigue). We suggest that this process is underpinned by long-term effects (unrecoverable) and short-term recoverable effects. In this decision making task any fatigue states were hidden, and could only be inferred by examining the changes in decisions across the experiment, where we indeed identify two different states (which we call fatigue) that influence choices on different timescales. From our data we cannot directly claim the source of these two different components of fatigue effect on motivation, but aim to in future work. In the revised manuscript we clarify this definition.
- b) There were a number of measures taken to ensure that the effects were not being driven by pain perception. Firstly, to avoid pain being the main driver of results the grip force devices were padded to avoid painful sensations in the hand. Secondly, the levels of force participants were exerting were very low in this experiment, with the highest force level in the main session only 48% of their maximum grip strength. These findings do not rule out pain sensations as driving our results, however, previous data examining muscle physiology and pain perception has shown that pain is not the driver of stopping behaviours in physical exertion and does not build up to any significant degree at low levels of force, even when participants continuously squeeze for extended periods of time (Staiano et al., 2018; Marcora et al., 2009). For these reasons we feel it is unlikely that sensations of pain are what is driving the results. However, even if it were pain sensations that were driving the results, our results would still suggest that short-term

and long-term sensations are leading to fluctuations in the willingness to exert effort for reward, which would still render the findings novel. In the revised manuscript we have highlighted these points

- c) To more directly test whether the results are reflecting people's representations of sensations of fatigue, we have now also included data from an additional experiment. In this additional experiment, participants performed a very similar task, except on each trial they were forced to either work or rest at a level of effort rather than being able to choose to do so. On each trial after working or resting, they also received a reward. The effort levels and reward levels were identical to those used in the fMRI experiment. Crucially on each trial they rated their level of tiredness (described to them as fatigue or tiredness) between zero and 100. We then fitted the five models used in the original manuscript to explain changes in trial-to-trial ratings in the new dataset. Replicating the original choice data, using Bayesian model comparison we show that the full model containing independent UF and RF components is the best explanation of fatigue ratings in this new data.

- d) Moreover, in the original dataset participants were required to rate from 0 (not at all) to 10 (extremely) how tired they felt before the start of the main task and after completion of the main task. We found that ratings of fatigue were increased at the end of the experiment compared to the beginning ($Z = 3.35$, $p < 0.001$) and that participants UF parameters correlated with the change in their fatigue ratings ($r_s(33) = .3614$, $p = .0329$, two-tailed). This supports the notion that sensations of fatigue were changing across the experiment and were linked to changes in their motivation as estimated by the computational model. These results have now been included in the manuscript

- e) Are the effects driven by accumulated reward reducing the marginal value of rewards? Theoretically it is unlikely that reward "satiety" in the manner suggested by the reviewer would be enough to cause people to switch away from working regularly in this experiment as there is limited evidence that people become hugely satiated to financial rewards in such short time frames. The smallest reward for working in this experiment is 6 times the magnitude of the rest option and thus would require a significant devaluation effect to cause a change in preference. To test empirically in the existing data is challenging, as participants choose to work less often when the offered effort is high and the reward is low. As such, accumulated effort and accumulated reward become correlated across the course of the experiment. However, in the additional experiment – where participants rate their fatigue – we can directly test if sensations of fatigue depend on the effort just exerted and reward that has been received as they are orthogonal. In the additional data, we find no evidence that ratings of fatigue are impacted by reward magnitude. Instead fatigue ratings depend largely on the effort exerted and can be captured effectively by the same computational model that was fitted to choice data. This provides strong evidence that our model is tracking the effects of fatigue induced by effort. Moreover, existing research examining the neural mechanisms consistently finds cumulative reward effects in VMPFC, but not in the regions we identified in this paper (Juchems et al., 2017; San Galli et al., 2018). Although

it is possible that accumulated reward also has some influence on choice, our results provide evidence that our computational model is tracking sensations of fatigue, that these influence choice and that it thus is likely to be identifying signals related to such sensations in the fMRI data.

All of these additional data and analyses are included in the revised manuscript in addition to multiple changes to the methods and results sections.

Page 7

In addition, we required participants to rate their level of “tiredness” – a more commonly used synonym of fatigue – between 0 and 10 before the start of the main task, and then again after completion of the experiment. Although participants could freely choose to rest, and thus prevent a significant build-up of fatigue, ratings of fatigue were higher at the end of the experiment than at the beginning ($t(34) = 4.27$, two-tailed $p < .001$, Cohen’s $d = 0.72$, 95% CI = [0.54, 1.52]; Supplementary figure. 2).

Page 22

It was beyond the scope of this investigation to examine whether the different components of motivational fatigue map onto purely psychological changes, physiological or metabolic changes in the state of the body, or fluctuations in neuromodulatory systems^{14,18,59}. However, the computational approach taken here was able to best explain changes in decisions about whether to exert effort for reward, and in self-reported sensations of fatigue. Although accumulated reward and accumulated effort were correlated in the fMRI study, rewards did not influence fatigue ratings trial by trial in the behavioural study. Such findings that sensations of fatigue were fluctuating in the experiment and could be quantified using the computational model in which effort exerted causes changes in motivation, rather than the rewards that have been accrued. In line with this, the UF and RF components fluctuated in regions that have previously been linked to effort processing, rather than in regions that have been found to signal accumulated reward (Juchems et al., 2017 & San Galli et al., 2018). Future work will need to identify the source of these fluctuating, putative fatigue states, and disentangle them from other processes, such as opportunity cost processing, boredom and time-on-task.

Supplementary Methods page 1

Grip force was measured using an MRI compatible, handheld dynamometer (TSD121B-MRI; BIOPAC Systems, Inc., USA) with padded “squash” tape to reduce discomfort.

Page 27

The effort levels used in this experiment were chosen as they have been shown not to cause significant build-up of lactate and muscle pain, and the stopping of exertion is driven more by the perception of effort and not pain. This ensured that our results are unlikely to be due to muscle pain, which can be incurred at higher levels of grip force.

Supplementary materials page 7

These results support the hypothesis that the value ascribed to exerting effort for reward fluctuates over time as a function of 'fatigue' as estimated by the model. However, there are other factors that may correlate with the effects of fatigue within this model, such as boredom or the accumulation of reward. To further demonstrate that this model was capturing sensations of fatigue, we correlated the model parameters for each participant with the change in their subjective ratings of fatigue before and after the main task. We found a significant correlation between the UF parameter (θ) and the change in rating ($r_s = .3614$, two-tailed $p = .033$, 95% CI = [0.032, 0.620]). Participants showing a greater increase in ratings of fatigue had a higher parameter weight, suggesting a greater reduction in the willingness to exert effort for reward due to UF. No significant correlations were identified between the parameters weights defining RF and the change in ratings, although such a result is to be expected as RF putatively only has short-term effects but ratings were taken more than one hour apart.

Main Text page 12

To further examine whether the computational model was able to capture sensations of fatigue, we performed an additional, similar behavioural experiment ($n=40$). In this study, participants performed a task with identical effort (0, 30, 39, 48%) and reward levels (6, 8, or 10 credits). However, rather than being able to freely choose whether to work or rest on each trial, instead they were required to exert a level of effort (or take a rest) and then rate their level of "tiredness" (a synonym for fatigue) on each trial. The computational model would predict that fatigue ratings would (i) increase as a function of effort exerted, (ii) would decrease after a trial of rest, (iii) the build-up would be best characterised by both RF and UF factors and (iv) would change independently of reward. In line with the predictions of our model, we found a significant effect of effort on trial-by-trial changes in fatigue ratings, a significant reduction in ratings after a trial of rest, but no significant effect of reward on ratings. To directly test these claims, we fit the five models that aim to capture changes in fatigue above to trial-by-trial ratings (Supplementary Methods & Results). The full model, containing separate RF and UF parameters better explained ratings than the other models. These results support the notion that our model is able to capture trial-by-trial changes in fatigue induced by effort, and its effects on the value ascribed to exerting effort for reward.

Supplementary Methods

Behavioural fatigue rating experiment

40 participants (24 females), mean age of 25.53 years ($SD = 5.63$; range 18-40) were tested on behavioural task in which participants exerted effort to receive rewards, and rated their level of tiredness on each trial. Unlike the fMRI experiment they were not able to choose whether to exert the effort or not, instead they required to exert (or rest) a level of force to receive rewards. After execution and receiving the reward they were required to rate their level of tiredness (supplementary Figure 5).

The experiment consisted of three parts: i) a *Calibration* phase to account for individual differences in strength, which was completed before the experiment was explained in full to the participants, ii) a *Training* phase in which participants familiarised themselves with the effort levels used in this task, and iii) the *Main task*. In the Main task, participants were asked on every trial to rest or to exert force for rewards (credits). Effort and reward levels were identical to the ones used in the main task of the fMRI experiment. Participants were instructed to collect as many credits as they could throughout the experiment, with the total number of credits collected throughout the task determining their payment. That is, participants were paid £8 for their time and received a bonus payment of up to £4 which was proportional to the credits they had earned in the task.

During *Calibration*, each participant's MVC was measured by squeezing a hand-held dynamometer on three consecutive trials with their dominant hand. Participants were required to apply as much force as possible on each trial, and they received strong verbal encouragement while squeezing. During each attempt, a bar presented on the screen provided feedback of the force being generated. In the second and third attempts, a benchmark representing 105% and 110%, respectively, of the previous best attempt was used to encourage participants to improve on their score. The maximum level of force generated throughout the three attempts was used as MVC.

In the *Training* phase participants practiced reaching each of four effort levels (0, 30, 39, and 48% of each participant's MVC). The trial was successful only when the force generated by the participant exceeded the required level for a sum total of at least 3 seconds in a five-second window. Each trial commenced with a cue in the form of a pie chart, with the number of red segments indicating the upcoming effort level. To make sure that participants carefully and successfully completed this training, they were awarded one credit for each successful squeeze, while they received zero credits for a failure. In an additional four trials, participants practiced manipulating the rating scale before they completed four full practice trials consisting of the different effort levels and a rating in order to familiarise themselves with the task.

The *Main task* (Supplementary Figure 5) consisted of 120 trials, each requiring participants to either rest or work for credits. Work trials consisted of one of three different effort levels, represented by two to four filled segments in a pie chart (cue) that corresponded to 30, 39, and 48% of each participant's MVC. Rest trials were indicated by one filled segment in a pie chart. The cue only indicated the effort level and not the reward. Rewards were presented for 1.5 seconds and only shown to the participants after they had worked or rested on that trial. Effort and reward levels were varied independently and presented in a pseudo-random order to ensure that 10 repetitions of each effort/reward combination was distributed evenly across the task, and each participant was presented with the same sequence to ensure that any potential differences in behaviour could be attributed to individual characteristics.

After this cue, participants were required to rest or to exert the respective force on the dynamometer for at least 3 out of 5 seconds in order to receive the credits. For this

purpose, participants were presented with a vertical bar that provided them with real-time feedback on their force. The target effort level was indicated by a yellow line superimposed on the bar. If participants had to rest on that trial, the bar was presented for the same duration but with the yellow line displayed at the bottom of the bar. Following this, participants were shown the credits they had obtained dependent on their success or failure on that trial. They then were asked to indicate how tired they felt on a scale ranging from 0 to 100, with 0 representing “not tired at all” and 100 representing “completely exhausted”. Immediately before the first trial, participants were given as much time as they needed to indicate how tired they currently felt. On each subsequent trial, the starting value on the scale was the value the participant had entered on the previous trial, and participants had a maximum of 5 seconds to either confirm or change this value. Participants could change the value on the rating scale in increments of 1 by using the left and right arrow keys on a keyboard. They then confirmed their chosen value by pressing the downward arrow key, and a green frame appeared around the rating scale. To ensure that participants reported their feelings of exhaustion accurately, it was made clear to them that none of their ratings would have an effect on the task they were asked to complete.

Fatigue rating experiment analysis

The main aim of this behavioural experiment was to examine whether fatigue ratings would be susceptible to the same short-term recoverable and long-term unrecoverable factors that were found to influence the choice data in the fMRI experiment. To test this, changes in fatigue ratings from trial $n-1$ to trial n were analysed with linear regression models fitted to each participant’s ratings, with predictors of effort, reward and their interaction.

Trials in which participants worked or rested were examined in separate LMMs than trials in which participants rested. This was due to the fact that “effort” is not a continuous variable in this experiment, with 0% effort not continuous between 30-50% force levels. Only trial n in which participants had successfully squeezed and thus obtained the credits were included in the model. This resulted in the exclusion of $M = 4.21\%$ ($SD = 6.60$) trials. Analysis at the group level was made by performing t-tests of normalised t-values against zero.

Modelling trial by trial fatigue ratings

To test whether the computational model fitted to choices in the fMRI experiment could also explain changes in fatigue ratings induced by effort and rest, we fitted the five computational models that predicted fatigue effects to the ratings data (Supplementary Figure X). On each trial t , fatigue (F) was calculated as the sum of a participant’s pre-task fatigue rating (F_{start}), recoverable fatigue (RF) and unrecoverable fatigue (UF):

$$F_{(t)} = F_{start} + RF_{(t)} + UF_{(t)} \quad (1)$$

Initial RF and UF values were set to 0, with RF and UF subsequently updated on each trial. RF increases dependent on the force (E) exerted on a trial (Equation 2) and decreases dependent on the time rested (T) on a trial (Equation 3):

$$RF_{(t)} = RF_{(t-1)} + (\alpha * E_{(t)}) \quad (2)$$

$$RF_{(t)} = RF_{(t-1)} - (\delta * T_{(t)}) \quad (3)$$

Individuals differ in the degree to which effort increases their fatigue, as reflected by the subject-specific parameter α , and in how quickly they recover during rest, reflected by the parameter δ . Unlike RF, UF accumulates depending on the effort exerted across the whole task and is not restored by resting during a trial (Equation 4). The parameter θ represents how quickly different individuals build up fatigue that cannot be easily recovered.

$$UF_{(t)} = UF_{(t-1)} + (\theta * E_{(t)}) \quad (4)$$

They were updated for every trial according to the model and added to the fatigue level indicated by the respective participant before the start of the task.

The fit between the model and the data, as indexed by the sum of squared residuals between the participant's ratings and the model's estimates, was optimised using *fminsearch* function in Matlab, i.e. model parameters were changed to minimise the difference between each participant's actual fatigue rating and the fatigue rating predicted by the model for each trial. To maximise the chances of finding global rather than local minima, parameter estimation for the full model and for all alternative models was repeated over a grid of initialisation values, with 6 initialisations (ranging from 0 to 1) per parameter. The optimal set of parameters for each model was used for model comparison.

To verify whether the three parameters used to quantify the effects of effort and rest were necessary, four other models were also fitted to participants' ratings. The alternative models either assumed (i) a UF component only (i.e. θ being fitted), (ii) an RF component only including one parameter, assuming that the rates of fatigue build-up during effort and of fatigue decrease during rest are similar, (iii) an RF component including one parameter as well as a UF component, and (iv) an RF component including separate work and rest parameters (i.e. α and δ being fitted) but no UF. In order to investigate the models' relative ability to predict the behavioural data, model fits were compared using Akaike Information Criterion (AIC) and Bayesian Information Criterion (BIC) with lower values indicating better fit.

Supplementary Figure 5. Behavioural study trial structure. Participants were required to exert force for rewards, with effort levels calibrated to their MVC (A), after training at each of four effort levels (0%,30,39,48% MVC) they performed 120 forced execution trails. On each trail they were instructed an effort that would be required (indicated by a pie chart), and then required to exert that level of force for a total of 3 out of 5s to obtain credits. They were then told the amount of credits received – 6, 8 or 10 credits if successful or 0 credits if failing to exert the required force. Following this they rated their level of tiredness from 0-100 on a continuous scale.

C) Significant effect of effort but not reward on fatigue ratings

Supplementary Figure 6. Behavioural study rating results. (A) Mean trial by trial ratings of “tiredness” between 0-100 across trials. Shaded areas represents SEM. (B) Change in fatigue ratings from t to $t-1$ as a function of effort level (x-axis) and reward (shade of blue) on trial t (i.e. the effort just exerted and the reward received for it). Only successful trials are included. Rest trials (0% MVC) induced recovery, with a linear effect of effort on increasing ratings. (C) Linear regression on trial to trial changes in ratings revealed significant effects of effort, but no significant effect of reward or effort x reward interaction. Error bars depict SEM. Results support the notion of a gradual increase in fatigue (A) across the experiment as well as trial by trial short-term recoverable changes (B).

A) Model comparison reveals best fit by full model

A) Full model is most frequently observed in the population

C) Models tested

Model 1: UF only (no recovery)

$$F_{(t)} = UF_{(t-1)} + \theta * E_{(t)}$$

Model 2: RF only, one parameter

$$F_{(t)} = RF_{(t-1)} + \alpha * E_{(t)} - \alpha * T_{(t)}$$

Model 3: RF, one parameter, and UF

$$F_{(t)} = (RF_{(t-1)} + \alpha * E_{(t)} - \alpha * T_{(t)}) + (UF_{(t-1)} + \theta * E_{(t)})$$

Model 4: RF only, two parameters

$$F_{(t)} = RF_{(t-1)} + \alpha * E_{(t)} - \delta * T_{(t)}$$

Model 5: Full model

$$F_{(t)} = (RF_{(t-1)} + \alpha * E_{(t)} - \delta * T_{(t)}) + (UF_{(t-1)} + \theta * E_{(t)})$$

Supplementary Figure 7. Computational modelling results for behavioural rating study. (A) Model comparison results in AIC for the models predicting changes in fatigue ratings across the experiment. The full model (5) is the best fit to the data when punishing for the number of parameters. (B) Exceedance probabilities for the models fitted to the ratings data Y-axis reflects the probability of being the most frequently observed model in the population. Model 5 is the winning “full” model of fatigue containing separate RF and UF components.(C) Models compared. All models predicted changes in fatigue ratings. The best fitting model predicted changes in fatigue that were partially recoverable - increase through effort, decrease through rest – but also contained a long-term unrecoverable component.

2) The evidence in favor of model 5 (with both RF and UF) compared to model 4 (RF only) is weak (slight difference in AIC or BIC). Besides, it seems that, as they are written in Fig. 3, model 5 and model 4 are no different. They can both be rewritten as $F(t) = \text{param1} \times \text{sum}(E) + \text{param2} \times \text{sum}(T)$, i.e. some weighted sum of cumulative effort and cumulative resting time. I might have missed something here, but at the very least this needs clarification.

Response: We thank the reviewer for raising these points about the extent to which this model successfully captures behaviour above alternatives. From a conceptual point we note that the models all try and capture a very similar phenomena – that is five of the models try and capture an increase in fatigue after effort and reduction after rest, and all of the models assume that rewards act as an incentive, increasing value, and the amount of force required serves as a cost decreasing it. It is therefore expected that the winning model would only show a small improvement over others. However, the two models are distinct. The full model contains separate weighting (i.e. a distinct parameter) on the effect of effort increasing RF from the degree to which effort increases UF. As such, people’s long-term increase in fatigue due to effort can be independent from the short-term increase in fatigue by effort. In the simpler model 4, this is not the case, there is only a singular parameter that dictates an increase in fatigue by effort. We have gone through the manuscript to ensure these points are clear.

However, we agree with the reviewer that it is important for our aims that we show that our model comparison is robust, that UF and RF are needed to explain the effects of fatigue and that our full model is the best characterisation of people’s behaviour. To address the reviewer’s concerns we have now (i) calculated exceedance probabilities (EP) on the AIC values (multiplied by -1) for the model comparison. EPs take a random-effects approach to calculate the probability that a winning model is the most frequent in the population. We find that the winning model has an EP of 0.72, and has a much higher probability than the next best model without UF which has an EP of 0.16. These EPs are now plotted in supplementary figure 4. (ii) We then fitted the five models that contain a change in fatigue used in the original manuscript to explain trial to trial ratings in the new dataset outlined above. Replicating the original choice data, using Bayesian model comparison we show that the full model containing independent UF and RF components is the best explanation of fatigue ratings in this new data. This demonstrates robustly that fatigue is made up of both a short-term and a long-term component and highlights that our fMRI results are likely correlating with such findings.

In the revised manuscript we include this additional behavioural experiment and the analyses of it in the supplementary material (details also included in reply to the last point), as well as including the exceedance probabilities from the original dataset:

Supplementary Figure 4. Supplementary results from fMRI study behaviour. (A) Histogram of proportion of trials on which participants made choices to “rest” out of 210 trials. There is considerable variability in choices both within and between many participants. (B) Exceedance probabilities for the models fitted to the choice data. Exceedance probabilities for models fitted to fMRI behavioural data. Y-axis reflects the probability of being the most frequently observed model in the population. The full model (5) is the most frequently best fitting model in the population. (C) Model parameters for each participant (green dot) and average across participants for the discounting parameter fitted to the pre-task (left), the two recoverable parameters (middle) and the unrecoverable parameter (right). Error bars represent SEM.

3) Simpler accounts must be included in the model comparison. The change in the ranges of reward and effort levels, between pre-task and main task, might by itself induce a change in the subjective value function. One possibility is for instance regression to the mean: in both cases participants tend to accept rewards above the mean, or efforts below the mean, which could be mistaken as a fatigue effect. I am aware that choices denote a shift of preference within the main task, but the change in the range might still affect conclusions, because it is not accounted for in model comparison. To fix the issue, an intercept parameter, capturing the difference between tasks, could be introduced in fatigue models.

Response: We appreciate the reviewer's point that simpler models could, in theory, explain changes in behaviour from the pre-task to the main task, with participants simply applying a similar heuristic to the value space or simply shifting their valuation down in a 'range-dependent' manner. We first would note that our behavioural analyses are not consistent with this, with the willingness to work changing across trials of the main task. This was shown statistically in our regression models, through the model comparison results, and also in the original figure 2E, which highlights a reduced willingness to exert higher efforts at lower rewards that changes between the first and second half.

Further to this, we now also include a supplementary figure displaying the first and last 27 trials of the task (below). As can be seen, participants were accepting offers at a very high rate (>90%) at the beginning of the main task, with even the highest effort/lowest reward offer still being accepted 50% of the time. This is consistent with what the model would predict would happen at the start of the main task, where only the short-term recoverable fatigue component would be having an effect. It is also inconsistent with a significant change in valuation from the pre-task to the main task. As can be seen, for the last 27 trials, there has been a significant change in behaviour, with offers accepted a low proportion of the time except at the lowest effort /highest reward levels. This is all consistent with significant changes in behaviour across the task driven by increased discounting, and an effect of both short-term recoverable and long-term unrecoverable fatigue effects.

We agree with the reviewer that this should also be compared within the model space. For this reason, we include model 6 which models a change in valuation (i.e. a new discount parameter) between the pre-task and the main task. This model would be able to capture a regression to the mean effect, however, we found that this model could not explain choice behaviour better than several of the models that contained an effect of fatigue on choice. Second, we tried to include the model suggested by the reviewer, in which an intercept term was included to explain differences between the tasks. However, the model fitting failed to find a stable solution, with participants model parameters differing each time the model fitting procedure was conducted and high correlations were identified between parameters. This is consistent with 'over-parameterisation' and thus suggesting that this model is not able to appropriately capture choices.

In the revised manuscript we have clarified how our results support the notion of valuations changing across the main task, included the additional figures and highlighted in the discussion how the results cannot be explained by a range-dependent or regression to the mean effects on value between the pre and main task.

A) Fatigue ratings increase after main task

B) Effort discounting in early and late trials in main task

Supplementary Figure 2. (A) Fatigue ratings taken before and after the main task of the fMRI study. Each dot represents one subject and error bars reflect SEM. Fatigue was higher after the main task than before. (B) Shift in choices award from higher effort lower reward options in last 27 trials compared to first 27 trials. Shift does not occur for lowest effort highest reward offer, consistent with a shift in valuation not more random behaviour.

Main text page 9

Consistent with a dynamic change in the value of working, there was a significant interaction between effort and cumulative effort, as well as main effects of effort, reward and cumulative effort (cumulative effort x effort: $Z = -4.19, p < .001$; effort:

-4.35, $p < .001$; reward: $Z = 3.96$, $p < .001$; cumulative effort: $Z = -3.83$, $p < .001$). The three-way ($Z = -0.27$, $p = .79$) and the cumulative effort \times reward interactions ($Z = 1.40$, $p = .16$) were not significant. Importantly, offers that were considered as higher in value and were chosen to work for at a high proportion in the pre-task, became gradually less and less likely to be selected across trials in the main task (Fig.2; Supp. Fig.2).

Supplementary Results page 7

Could these results be due to participants becoming more random in the main task due to boredom or other confounding factors? We show that Effort and Reward still have very strong significant effects when examining only the last quarter of trials (Effort: $Z = -5.2319$; $p < .0001$; Reward: $Z = 4.3050$; $p < .0001$). As such participants were basing their behaviour strongly on the effort levels towards the end, which is not consistent with more random behaviour. In addition, participants were still choosing to work on almost 100% of trials for the highest reward and lowest effort in the last 27 trials of the main task (see supplementary figure 2), which is also inconsistent with fatigue causing more stochastic behaviour.

4) Besides, instead of cumulative effort, or cumulative reward (as suggested above), a simpler function of time-on task, like trial number, should be tested. It could simply be that participants are more and more bored with the task, or willing to go home. A way to show that fatigue is really about effort cost would be to sum quadratic (not linear) effort levels. Also, instead of an effect on effort cost (or reward benefit, as suggested above), trial index or fatigue could impact an additive parameter, which would suggest that they are just less willing to squeeze anymore, irrespective of reward and effort levels.

Response: The reviewer raises an important point that boredom could be a factor in people's choice behaviour in the main task. Although we cannot rule out this possibility, predictions about how choice behaviour changes across the experiment can be made in relation to boredom: (i) it would lead to more distracted behaviour which would result in more missed trials over time or more trials in which the required force is not exerted. However, in the experiment missed trials and "failed" comprise less than 2%, such that there is not enough variance across participants to statistically test if missed trials increased with trial number, suggesting that participants were not becoming less attentive to the task and (ii) choices that are more random and would not depend on the offered effort and reward levels would occur at the end of the experiment. However, statistically we show that Effort and Reward still have very strong significant effects when examining only the last quarter of trials (Effort: $Z = -5.2319$; $p < .0001$; Reward: $Z = 4.3050$; $p < .0001$) and showed a significant cumulative effort \times effort interaction ($p < 0.001$), with the effect of effort stronger at the end of the experiment than at the beginning. As such participants were basing their behaviour more strongly on the effort levels towards the end, which is not consistent with boredom leading to a generalised reduction in rejecting offers to work irrespective of effort and reward. In addition, participants are still choosing to work on almost 100% of trials for the

highest reward and lowest effort in the last 27 trials of the main task (see supplementary figure 2), which is also inconsistent with boredom leading to a rejection of all offers and a non-specific reduced willingness to squeeze at the end of the task.

In addition to this, we also note that in the new behavioural data from the task with trial-to-trial ratings, participants show changes in their ratings of fatigue across the whole task. Although self-report, this data corroborates sensations of fatigue impacting on motivation. As such, our results are consistent with the notion of sensations of fatigue changing that impact on the motivation to exert effort for reward, rather than boredom. In the revised manuscript we have highlighted these analyses and discuss whether the effects could be related to boredom rather than fatigue driving changes in motivation.

The reviewer's second point regarding the quadratic effect of effort on fatigue is an interesting one. However, there are two difficulties in estimating this in the main task (i) there are only three effort levels, as such a quadratic or linear effect of effort on fatigue cannot easily be distinguished as there is not enough parametric range to distinguish between them, (ii) the model is estimated on choice data, where the valuation is parabolic, as has been shown repeatedly for effort-based decision-making (Chong et al., 2017, Lockwood et al., 2017). As a result, an increase in fatigue causes a parabolic effect on choice behaviour. Models with linear increases or squared increases in fatigue will likely not be distinguishable – other than with small changes in parameter weights. However, in the additional data of trial-by-trial ratings we show changes in ratings of fatigue are linear in form rather than quadratic, suggesting that for the effort levels used in this task the effects of effort on fatigue are linear (see Supplementary Figure 6). In the revised manuscript we now highlight these points in the discussion as well as in the supplementary results.

Main text page 10

Importantly, offers that were considered as higher in value and were chosen to work for at a high proportion in the pre-task, became gradually less and less likely to be selected across trials in the main task with effort and reward having stronger effects in even in the last 27 trials of the experiment (Fig.2; Supp. Fig.2). Such findings are inconsistent with boredom or other factors leading to generally more noisy or random behaviour. Instead these results are indicative of participants changing their subjective evaluation of working across trials, with accumulated efforts increasing the discounting effect of effort and reducing the value of working across the experiment.

Main Text page 19

Although in this study we cannot fully rule out the possibility that this effect is simply due to time-on-task or boredom effects, we are able to show that it affects both effort-based decisions and self-reported fatigue. Moreover, we show that this longer-term effect is independent from a short-term recoverable component and that it covaries with activity during effort-based choice in the left MFG, largely overlapping with an area which has previously been associated with subjective aversion to cognitive effort ⁴⁰.

Main Text page21

It was beyond the scope of this investigation to examine whether the different components of motivational fatigue map onto purely psychological changes, physiological or metabolic changes in the state of the body, or fluctuations in neuromodulatory systems^{14,18,61}. However, the computational approach taken here was able to best explain changes in decisions about whether to exert effort for reward, and in self-reported sensations of fatigue. Although accumulated reward and accumulated effort were correlated in the fMRI study, rewards did not influence fatigue ratings trial by trial in the behavioural study. Such findings, that sensations of fatigue were fluctuating in the experiment and could be quantified using the same computational model in which effort exerted causes changes in fatigue, suggest that changes to choice behaviour in the fMRI experiment are more likely to be due to exerted effort than accrued reward. In line with this, the UF and RF components fluctuated in regions that have previously been linked to effort processing, rather than in regions that have been found to signal accumulated reward^{62,63}. Future work will need to identify the source of these fluctuating, putative fatigue states, and disentangle them from other processes, such as opportunity cost processing, boredom, task switching and time-on-task.

Neuroimaging (claims 2 and 3)

5) The finding that left MFG decreases with fatigue (be it RF or UF) is convincing, as the cluster shows up in a whole-brain analysis, surviving correction for multiple comparisons. However, its contribution to the shift in preference could be specified. What the analysis shows is that its activity is decreasing with time on task (or fatigue), but the link to choices is not established. Could it be that left MFG is simply less active when effort is declined, which becomes more and more frequent across task trials? Would this be related to shorter deliberation time?

Response: We thank the reviewer for their question regarding the role of the MFG in the choice components. Its important to note that our results do support the role of the MFG in the decision process, by influencing the degree to which a reward is devalued by effort. In this regard, it is linked to choice behaviour. It is plausible that MFG activity simply decreases as offers are rejected, however, our results are more consistent with the findings of Blain and colleagues, who implicate this region in “executive fatigue” in healthy people and over-training in athletes. Future work will need to dissect its contribution to changes in decision processes linked to fatigue. In the revised manuscript we highlight how the MFG could contribute to decision processes and also include an RT analysis

Main Text page 19

This study unifies separate lines of research that have theorised that the effects of fatigue may occur on more than one timescale^{14,35}, and provides a formalised account of their effects on effort-based decisions. One line of research had suggested that extended periods of work lead to exhaustion that has consequences for tasks performed after the

one which caused the fatigue ^{11,17,36–38}. This “executive fatigue” influences activity in the MFG in tasks performed after having been exhausted, an effect exacerbated in athletes who are over-trained ^{11,39}. This form of fatigue appears to be unrecoverable in the sense that simply taking short rests does not have a restorative effect. Although in this study we cannot fully rule out the possibility that this effect is simply due to time-on-task or boredom effects, we are able to show that it affects both effort-based decisions and self-reported fatigue. Moreover, we show that this longer-term effect is independent from a short-term recoverable component and that it covaries with activity during effort-based choice in the left MFG, largely overlapping with an area which has previously been associated with subjective aversion to cognitive effort ⁴⁰.

Page 35

In addition, to examine activity that covaried with reaction times, we also ran a separate GLM in which only individual trial by trial reaction times were included as a regressor (Supplementary table 4).

Supplementary Table 4

Anatomical locations in which activity significantly covaried with individual reaction times at $p < .001$, uncorrected for multiple comparisons

Anatomical area	MNI peak	No. of voxels	Z-value	Voxel $p_{\text{uncorr.}}$
Left anterior insula	-33, 17, 5	11158	7.69	< 0.001*
Brain stem	6, -22, -7	367	6.26	< 0.001*
Left inferior temporal gyrus	-45, -58, -13	338	5.32	< 0.001*
Right pallidum	18, 2, -1	82	4.31	< 0.001
Left cerebellum	-36, -52, -34	32	4.28	< 0.001
Right inferior temporal gyrus	48, -61, -13	154	4.23	< 0.001
Brain stem	6, -28, -34	68	4.17	< 0.001
Left medial orbital gyrus	-21, 38, -22	13	3.99	< 0.001
Left medial orbital gyrus	-15, 11, -19	8	3.61	< 0.001
Left thalamus	-12, -22, 14	8	3.55	< 0.001
Right frontal pole	21, 65, -1	6	3.52	< 0.001
Left hippocampus	-24, -25, -4	6	3.44	< 0.001
Right central operculum	48, -19, 17	8	3.32	< 0.001
Right inferior frontal gyrus	51, 47, -10	7	3.29	< 0.001

** indicates significance at a threshold of $p < .05$ with a whole-brain voxel-level family-wise error correction.*

6) On the contrary, the dissociation between RF and UF relates to cingulate zones that do not appear in activation tables, even at uncorrected threshold. They only survive small-volume correction within pre-defined regions of interest that seem quite arbitrary (why

not other regions, like the anterior insula?). I think this level of evidence is way too weak to maintain a conclusion such as neural correlates of RF and UF can be dissociated.

Response: The reviewer's points about significance are important, and allow us to clarify our analyses, to highlight that our results are reported at a stringent threshold correcting for multiple comparisons but also test anatomically specific hypotheses.

In regards to the definition of regions we tested specific hypotheses about two Cingulate/SMA zones which have been implicated consistently in studies in effort-based decision-making and research on fatigue across species and methodologies (Klein-Flugge et al., 2015, Chong et al., 2017, See Muller et al., 2019; Le Heron et al., 2018; Vassena et al., 2019; Verguts et al., 2015 for reviews; see Pessiglione et al., 2018 for a meta-analysis). Of all the regions in the brain, this medial frontal cortex region and the VS are the most commonly implicated regions. Moreover, these regions have also been implicated in studies examining how value-based decisions change across contexts and timescales (Kolling et al., 2016; Wittmann et al., 2018). As such, we had considerable evidence to test hypotheses about the contributions of the medial frontal cortex to how effort-based decisions change across trials in this experiment.

It is also important to note that our approach to correcting for multiple comparisons with volume-based corrections rather than whole brain is a common approach to controlling for false positives in fMRI data. However, our particular approach is more statistically conservative than is common in studies that take a ROI based approach. To define the volume used for correction for multiple comparisons we merged together the different masks that comprised our hypothesised regions, and thus the statistical threshold used for voxels was defined by the number of voxels across masks, not only for a particular cingulate zone. This means that a more stringent threshold is imposed, with a larger number of voxels being corrected for than that which are in the region of interest. However, we are able to benefit from the fact that individual masks for the different zones were anatomically precise, and thus allowed us to localise our results for UF and RF to distinct cingulate ones. Thus, we controlled for false positives across regions, as well as within regions, but were also able to make more precise anatomical arguments. This approach is much more statistically conservative, and sets a more stringent statistical threshold than approaches often used when testing hypothesised regions i.e. only correcting for a single small-volume or by only correcting regions by the number of ROIs included in the analysis (not their number of voxels). It is for this reason that the Z-values for all of our main results are very high, with the *least* significant key result having a Z-score of 3.57 – corresponding to a p-value of 0.0001. In debates around statistical thresholding in fMRI, Z-scores ranging between 2.5-3.1 are the most commonly deployed (Woo et al., 2014) and this result is therefore clearly far in exceedance of such thresholds and that includes in results published very recently in this journal (e.g Fromer et al., 2019; Hogan et al., 2020). As such, although we have used a “small” volume correction the results reached a statistical threshold far more stringent than those reported in many papers.

In terms of the Insula, we had not focused on this region, as although there is some evidence linking this region to effort and fatigue, it is more mixed than that of the medial frontal cortex – and loci differ across studies from very posterior regions (Meyniel et al., 2013) to more anterior (Chong et al., 2017). However, we appreciate that readers may be interested in results in other regions, including the insula. Interestingly, we found that activity in the anterior covaried positively with the unrecoverable fatigue. We have now highlighted this additional result in the manuscript.

In regards to the presence of these results in uncorrected tables, the UF is indeed present in supplementary table 1, in addition RF is present in the table, but was part of a larger cluster with multiple peaks. This is common when examining results at reduced thresholds, where independent clusters end up appearing joined together due to voxels at their extremities showing only weak significance that is present at reduced thresholds. However, as noted above, the result correspond to a highly significant effect, even if its peak is not apparent in the table.

In the revised manuscript we more clearly highlight our strategy, that our results are significant at very stringent thresholds and highlight that the results are not present in the tables at reduced threshold due to merging with other clusters.

Main Text Page 35

Because previous studies have emphasised the importance of the VS and the dACC/pre-SMA region in processing effort-based decisions, and in order to be able to specifically localise activity to anatomically and functionally distinct regions, we also probed these areas using a priori regions of interest. Therefore, t-contrasts were conducted at the whole-brain level at an uncorrected statistical threshold of $p < .001$, and then a FWE small volume correction was applied using a combined mask taken from appropriate atlases (bilateral VS: from Harvard-Oxford Atlas; bilateral dACC and pre-SMA: areas RCZa, RCZp and pre-SMA defined through resting-state parcellations of the frontal cortex by Neubert and colleagues⁴¹). By combining these masks together we provide a more conservative statistical threshold than individual ROI analyses, balancing possible false negatives that can occur with whole-brain correction. Full tables of results are reported at an uncorrected threshold of $p < .001$ in Supplementary tables 1 to 3. At this reduced threshold clusters in the VS and RCZp lie within a larger cluster and thus do not show in the list of peak results only.

7) There is a double-dipping issue when selecting clusters based on regression against RF or UF and then comparing regression estimates extracted from the peak of these clusters. The issue is that the selection is not independent from the comparison, meaning that it is biased towards voxels in which the noise will favor a significant comparison.

Response: We appreciate the important point regarding avoiding double-dipping in analysing fMRI data. The reviewer's comment gives us the opportunity to highlight that we did not define our regions of interest, or make statistical inferences, based on comparisons between regression estimates extracted from the peak results. The values shown in the figures are extracted from the peak voxels, for display purposes, but statistical inferences were made in the following manner to avoid double-dipping: (i) To first examine whether regions showed significant effects of RF, UF or SV, we examined whether we could identify voxels within the hypothesised regions that significantly covaried with each of these parameters separately, (ii) to test if the same region also encoded one of RF, UF or SV, more than the other variables, we performed independent contrasts between each of these variables (RF>UF, RF>SV etc.) and examined whether there were voxels present in the same anatomical zone that overlapped with voxels defined by the results from (i). Thus, we interpret results based on significant overlapping voxels within a specific anatomical zone across these analyses, rather than from values extracted from a region. Thus, we have not taken approaches that would have led to double-dipping. In the revised manuscript we highlight this approach and highlight how we avoid our results being driven by false positives due to double-dipping.

Main Text page 14

We therefore examined voxels in which activity at the whole brain level and within our hypothesised regions of interest ([ROI] – See Methods) significantly covaried with RF or UF. Then we tested whether the same voxels significantly covaried with one parametric regressor and did not significantly covary with the others. Such an approach of examining overlap avoids the problems of double-dipping in ROI based analyses.

8) Showing that activity in neural regions like the ventral striatum correlates with fatigue-weighted subjective value is no proof that fatigue does affect value signals in these regions. This is because subjective value integrates factors (reward and effort levels) that are sufficient to explain the correlation. In other words, VS activity might correlate with fatigue-weighted SV just because it responds to rewards. To prove their point, the authors need to show that neural activity in VS or other regions is better explained by fatigue-weighted SV than by regular SV (without fatigue).

Response: We thank the reviewer for giving us the opportunity to highlight our reasoning for considering signals in the VS and SFG as signalling value, weighted by fatigue, rather than a consistent value signal across the experiment. The reviewer is correct that we have not statistically shown that signals in these regions covary with fatigue-weighted value per se. Such a comparison is challenging because both fatigue-weighted value and the subjective value without an influence of fatigue are highly correlated (0.7) – this is unsurprising as both are defined by both reward and effort as they note. However, what could not be explained by a static SV computation, would be the signal within a region correlating with individual subjects' fatigue parameters from the computational model. In the VS we found that individual differences in signalling covaried with each of the fatigue defining parameters. This supports the notion that signals in the VS depended on a person's sensitivity to fatigue, which supports the idea that this region was signalling value, weighted by fatigue. In the

revised manuscript we highlight how we interpret the VS results and fatigue-weighted value and include points in the discussion about how we can interpret signals as value vs fatigue-weighted value.

Page 21

Moreover, they point to a role of the VS for integrating current levels of fatigue with the value of persisting with a demanding task, and variability between people in such tendencies. A limitation of the experiment is that comparing value to fatigue-weighted value signals is challenging, as they are necessarily correlated within the design. However, importantly, we found that variability in signalling in the VS between people correlated with the parameters of the computational model. Such a finding is consistent with activity in the VS signalling a dynamically changing estimate of value, which is weighted by each participants' tendency to persist in the face of momentary fatigue. Such effects would be missed or confounded by typical analysis approaches, e.g. when examining changes correlated with trial number or behaviour pre vs post exhaustion, but they can be examined using the formal framework outlined here. Future work will need to understand how the VS integrates fatigue and value-related information leading to fluctuations in the motivation to persist with ongoing behaviour.

9) There is the same difference between frontal pole and VS as between MFG and cingulate zones: the former activation is convincing because it survives whole-brain corrected threshold, while the latter rely on a priori ROI. However, I would question the 'frontal pole' label, which usually refers to BA 10. From the map on Fig. 5 it seems that the cluster is more dorsal and posterior, more like superior frontal gyrus (sometimes called dorsomedial prefrontal cortex).

Response: We thank the reviewer for noting that our label of frontal pole may not have been clear. The result extended across the areas 9 and 10 on the Superior Frontal Gyrus (SFG). We have now re-labelled these results where appropriate to clarify this.

In regards to the VS, this region has been one of the most commonly reported regions in studies of value-based decision-making, including those examining effort-based decisions across species (Kurniawan et al., 2010; Schuoppe et al., 2014; Massar et al., 2015; Salamone et al., 2007) and thus we had strong hypothesis linking its function to value processing in this task. Much like the result in the cingulate zones, although not bonferroni whole brain corrected (which is open to false negatives) the result still survives a high threshold of multiple comparison correction and has a Z-value of 4.3. In line with our previous comment, we highlight that this is a robust result that passes a stringent threshold, and is statistically stronger than results reported in many recently published papers.

In the revised manuscript, we expand our justification for the VS region of interest and highlight how this result is statistically robust and also have changed figure 5 and discussion of the superior frontal result to Frontal Pole / Superior Frontal Gyrus throughout.

Page 4

and ventral striatum (VS) have been implicated in computing value and motivating the exertion of effort^{5,6,8,26-32}. Evidence suggests that these regions also change their response with time on task^{14,16,18}

Minor issues:

- Introduction and discussion could more focused, at present there are many redundancies, and the links with cited papers are often loose. Also, the novelty of the computational framework is clearly oversold: increasing effort cost with cumulative effort is quite a standard solution.

Response:

We apologise to the reviewer for any lack of clarity in the Introduction and if any of the claims appeared oversold, this was not the intention. We merely wanted to highlight that there has yet to be a model of fatigue that unifies the short-term and long-term effects of fatigue and can measure these effects on trial by trial decisions of whether exerting effort for reward is 'worth it'. We have now revised the wording throughout to reflect this.

- Fig. 2D is not particularly useful, as I cannot see the fatigue effect (I presume the plot is meant to show darker choice probability with progress in the task).

Response: We apologise that the reviewer found this figure panel unhelpful. We believe its inclusion is useful to highlight that work/rest decisions fluctuate over the experiment, and relate to a comment by another reviewer, but also to highlight that *when* people make decisions to rest is not highly consistent – which is why having a model that accounts for variability between people is important. We have now clarified these points in the text and figure legend.

(d) Percentage of participants who accepted the work offer, illustrated separately for each trial in the main task. Values reflect the consistency with which trials were accepted or rejected across the experiment. This shows considerable variability in choices, but high levels of choices to “work” in early trials, rather than late.

- The analysis in Fig. 3C is at odds with the winning computational model, because RF and UF are now included as additive (significant) regressors, instead of interacting with effort cost. If the idea is to provide further evidence in favor of the model, this is not helpful. It rather suggests that choices are more and more biased towards rest with increasing trial number.

Response: The reviewer is correct that this analysis showed an effect of RF and UF independently of effort, however, we agree that it is more informative to show an interaction with effort. In the revised manuscript we have included an updated analysis showing significant interactions between effort level on a trial and UF and RF levels according to the model.

Page 12

To test that this model was not only better than the alternatives but also significantly predicted choice behaviour, we performed a logistic regression on work versus rest decisions including z-scored reward and z-scored interactions of effort and trial-by-trial model-estimated recoverable and unrecoverable fatigue as predictors. As in the previous logistic regressions, reward significantly predicted choice ($p < .001$), but crucially there were also significant negative interactions of effort and both fatigue components (recoverable fatigue \times effort: $Z = -4.98$, $p < .001$; unrecoverable fatigue \times effort: $Z = -5.17$, $p < .001$). This was the case whether using the average estimated fatigue across participants or the model's idiosyncratic estimate of fatigue from each participant (all $ps < .001$). Thus, when the levels of fatigue in the model were higher, it was predictive of a greater tendency to rest, in particular when higher effort levels were on offer. Therefore, the willingness to exert effort for reward is not static but fluctuates moment-to-moment. When fatigue states in our model are higher this is related to reduced motivation and crucially a greater discounting of reward by effort.

- I did not find any information about how participants were remunerated. This is important to discard the possibility that they simply trade their payoff against time on task, instead of squeezes.

Response: We apologise that this information was not included in the manuscript. Participants were paid a default of £25 for participating in the experiment and could earn up to £10 further as a bonus for the rewards collected during the experiment, which depended on the choices they made and successful exertions of required effort.

Page 24

Participants were remunerated with £25 for taking part in the study, plus a possible £10 further as a bonus payment. The bonus depended on the credits accrued on all successfully executed trials of the main task, as well as trials executed during the training and pre-task phase. Thus an increase in the bonus was an incentive on every trial.

- There are a few typos that need correction (e.g., Fig. 3B: "Schematic representation for how F ... effect value and choices to work or rest").

Response: Thank you for highlighting these typos, these have now been corrected.

Reviewer #3 (Remarks to the Author):

In this interesting manuscript the authors explore the role for trial-wise fatigue in the neural computations of effort-based decision-making. The authors present evidence for two distinct fatigue signals that are distributed across various nodes within a fronto-striatal network. Fatigue is an under-studied and poorly understood construct, and this work therefore has the potential to make a significant and innovative contribution. The paper is superbly written and the analytical methods are sophisticated and appropriate. Despite these strengths, I do have some (mostly minor) concerns with aspects of the analysis and some more significant concerns related to interpretation. I have the following comments for the authors to consider:

Response: We thank the reviewer for their positive evaluation of the work. Their comments have allowed us to improve the manuscript and include additional data that addresses their concerns.

The most significant problem as I see it is that “fatigue” is an under-specified construct both conceptually and operationally. Specifically, it seems likely that there are likely moderate to high correlations between the parameters representing recoverable fatigue (RF) and unrecoverable fatigue (UF) and other decision-variables. As I understand the task design and computational model, the UF parameter scales the cumulative expenditure of effort. However, it would seem that cumulative expenditure of effort would also be highly correlated with cumulative rewards, as the effortful option always yields greater rewards. Therefore, this parameter could capture diminishing marginal utility of accumulating points over the course of the task. It would also necessarily correlate at least moderately strongly with the mere passage of time. As such, the strict interpretation as a measure of “unrecoverable fatigue” seems hard to justify. One could just as easily think of it as a global “opportunity cost signal” reflecting the additive and/or interactive effects of fatigue, diminished interest in additional points, a desire to finish up the study and move on to other activities, etc., etc., Indeed, such global opportunity cost signals have been predicted in the context of effort (e.g., Kurzban et al., 2014).

Similarly, for the interpretation of the RF parameter as representing “recoverable fatigue”, other interpretations seem equally plausible. It would seem this value might also correspond with foraging values, task switching, etc., all of which could be consistent with the observed results in terms of both the computational model and the imaging results in the RCZ. The authors acknowledge this on the one-hand, but still claim that their work shows a unique RF contribution. But without ruling out the possibility that RF is merely tracking with other decision-variables, claiming a unique RF component seems to be an over-reach.

Response: The reviewer raises a number of important questions about how strongly our results can claim to be linked to “fatigue” versus other potentially correlated phenomena such as opportunity cost processing, or cumulative reward. The reviewer is correct that some of these questions were hard to address in the existing dataset. However, to more robustly test if our computational model could capture moment-to-moment changes in sensations of fatigue as well as changes in motivation from trial to trial we have included an additional experiment. In this additional data, participants performed a very similar experimental task, except on each trial they were forced to either work or rest rather than being able to choose to do so. Rather than making choices, on each trial they then rated how tired they were between zero and 100. We then fitted the five models used in the original manuscript to explain changes in trial by trial decisions, to explain trial to trial ratings in the new dataset. Replicating the original choice data, using Bayesian model comparison we show that the full model containing independent UF and RF components is the best explanation of fatigue ratings in this new data. This demonstrates robustly that people’s subjective report of fatigue is made up of both a short-term and a long-term component. In this additional dataset we also find no evidence that rewards influence trial by trial ratings of fatigue. Thus, although it is plausible that other decision variables may correlate with RF and UF, these results add weight to the notion that there are sensations of fatigue that build up over a short and longer timescale, that is not directly tied to decision variables other than the accumulated effect of exerting effort.

In the revised manuscript we have included the full results of this additional dataset and revised discussion where we note that our results could relate to other phenomena and decision variables, but also are consistent with subjective reports of fatigue.

Main Text Page 12

To further examine whether the computational model was able to capture sensations of fatigue, we performed an additional, similar behavioural experiment (n=40). In this study, participants performed a task with identical effort (0, 30, 39, 48%) and reward levels (6, 8, or 10 credits). However, rather than being able to freely choose whether to work or rest on each trial, instead they were required to exert a level of effort (or take a rest) and then rate their level of “tiredness” (a synonym for fatigue) on each trial. The computational model would predict that fatigue ratings would (i) increase as a function of effort exerted, (ii) would decrease after a trial of rest, (iii) the build-up would be best characterised by both RF and UF factors and (iv) would change independently of reward. In line with the predictions of our model, we found a significant effect of effort on trial by trial changes in fatigue ratings, a significant reduction in ratings after a trial of rest, but no significant effect of reward on ratings. To directly test these claims, we fit the five models that aim to capture changes in fatigue above to trial by trial ratings (Supplementary Methods & Results). The full model, containing separate RF and UF parameters better explained ratings than the other models. These results support the notion that our model is able to

capture trial by trial changes in fatigue induced by effort, and its effects on the value ascribed to exerting effort for reward.

Main Text page 22

It was beyond the scope of this investigation to examine whether the different components of motivational fatigue map onto purely psychological changes, physiological or metabolic changes in the state of the body, or fluctuations in neuromodulatory systems^{14,18,59}. However, the computational approach taken here was able to best explain changes in decisions about whether to exert effort for reward, and in self-reported sensations of fatigue. Although accumulated reward and effort were correlated in the fMRI study, rewards did not influence fatigue ratings trial by trial in the behavioural study. Such findings, that sensations of fatigue were fluctuating in the experiment and could be quantified using the same computational model in which effort exerted causes changes in fatigue, suggest that changes to choice behaviour in the fMRI experiment are more likely to be due to exerted effort than accrued reward. In line with this, the UF and RF components fluctuated in regions that have previously been linked to effort processing, rather than in regions that have been found to signal accumulated reward (Juchems et al., 2017 & San Galli et al., 2018). Future work will need to identify the source of these fluctuating, putative fatigue states, and disentangle them from other processes, such as opportunity cost processing, boredom, task switching and time-on-task.

Supplementary Figure 5. Behavioural study trial structure. Participants were required to exert force for rewards, with effort levels calibrated to their MVC (A), after training at each of four effort levels (0%,30,39,48% MVC) they performed 120 forced execution trails. On each trail they were instructed an effort that would be required (indicated by a pie chart), and then required to exert that level of force for a total of 3 out of 5s to obtain credits. They were then told the amount of credits received – 6, 8 or 10 credits if successful or 0 credits if failing to exert the required force. Following this they rated their level of tiredness from 0-100 on a continuous scale.

C) Significant effect of effort but not reward on fatigue ratings

Supplementary Figure 6. Behavioural study rating results. (A) Mean trial by trial ratings of “tiredness” between 0-100 across trials. Shaded areas represents SEM. (B) Change in fatigue ratings from t to $t-1$ as a function of effort level (x-axis) and reward (shade of blue) on trial t (i.e. the effort just exerted and the reward received for it). Only successful trials are included. Rest trials (0% MVC) induced recovery, with a linear effect of effort on increasing ratings. (C) Linear regression on trial to trial changes in ratings revealed significant effects of effort, but no significant effect of reward or effort \times reward interaction. Error bars depict SEM. Results support the notion of a gradual increase in fatigue (A) across the experiment as well as trial by trial short-term recoverable changes (B).

A) Model comparison reveals best fit by full model

A) Full model is most frequently observed in the population

C) Models tested

Model 1: UF only (no recovery)

$$F_{(t)} = UF_{(t-1)} + \theta * E_{(t)}$$

Model 2: RF only, one parameter

$$F_{(t)} = RF_{(t-1)} + \alpha * E_{(t)} - \alpha * T_{(t)}$$

Model 3: RF, one parameter, and UF

$$F_{(t)} = (RF_{(t-1)} + \alpha * E_{(t)} - \alpha * T_{(t)}) + (UF_{(t-1)} + \theta * E_{(t)})$$

Model 4: RF only, two parameters

$$F_{(t)} = RF_{(t-1)} + \alpha * E_{(t)} - \delta * T_{(t)}$$

Model 5: Full model

$$F_{(t)} = (RF_{(t-1)} + \alpha * E_{(t)} - \delta * T_{(t)}) + (UF_{(t-1)} + \theta * E_{(t)})$$

Supplementary Figure 7. Computational modelling results for behavioural rating study. (A) Model comparison results in AIC for the models predicting changes in fatigue ratings across the experiment. The full model (5) is the best fit to the data when punishing for the number of parameters. (B) Exceedance probabilities for the models fitted to the ratings data Y-axis reflects the probability of being the most frequently observed model in the population. Model 5 is the winning “full” model of fatigue containing separate RF and UF components. (C) Models compared. All models predicted changes in fatigue ratings. The best fitting model predicted changes in fatigue that were partially recoverable - increase through effort, decrease through rest – but also contained a long-term unrecoverable component.

Related to the above, it was a bit surprising not to see subjective report of fatigue and its association with model parameters. While self-reported fatigue has its own measurement limitations, it would nevertheless provide some additional evidence that the putative fatigue parameters are tracking with the subjective experience of fatigue. If these data were collected as part of this study then they should be included. If not, it could potentially be included in a follow-up behavioral study in a separate sample.

Response: We thank the reviewer for raising this point. In addition to the results now included from the additional experiment outlined above, which highlights that fatigue is being reported by participants in an almost identical experiment, we also now include ratings from the participants from the fMRI experiment taken from before and after the main task. We did not include this data originally, as it has significant limitations. Firstly, for practical reasons participants did not rate their sensations of fatigue immediately before and after the main task inside the MRI scanner – there were short breaks before and after. As such, short-term fluctuations in fatigue are unlikely to consistently influence ratings. Secondly, participants were free to choose whether to work or rest, and thus, by design had the opportunity to avoid becoming too fatigued. As such, these ratings should be interpreted with caution. However, the data does suggest that participants were being fatigued during the experiment and that this related to a component of the model. Participants were required to rate from 0 (not at all) to 10 (extremely) how tired they felt at the moment both before the start of the main task and after completion of the main task. We found that ratings of fatigue were increased at the end of the experiment compared to the beginning ($Z = 3.35$, $p < 0.05$) and that participants UF parameters correlated with the change in their fatigue ratings ($r_s(33) = .3614$, $p = .0329$, two-tailed). This supports the notion that changes in motivation evident in the choice behaviour of the fMRI participants were linked to changes in their levels of fatigue/tiredness. We did not find any correlation with other parameters, however, this is to be expected as the RF component models short-term effects of fatigue, which are unlikely to have an effect on participants ratings at the end of the experiment. In the revised manuscript we now include this data in supplementary materials and supplementary figure 2.

Page 12

These results support the hypothesis that the value ascribed to exerting effort for reward fluctuates over time as a function of ‘fatigue’ as estimated by the model. However, there are other factors that may correlate with the effects of fatigue within this model, such as boredom or the accumulation of reward. To further demonstrate that this model was capturing sensations of fatigue, we correlated the model parameters for each participant with the change in their subjective ratings of fatigue before and after the main task. We found a significant correlation between the UF parameter (θ) and the change in rating ($r_s = .3614$, two-tailed $p = .033$, 95% CI = [0.032, 0.620]). Participants showing a greater increase in ratings of fatigue had a higher parameter weight, suggesting a greater reduction in the willingness to exert effort for reward due to UF. No significant correlations were identified

between the parameter weights defining RF and the change in ratings, although such a result is to be expected as RF putatively only has short-term effects but ratings were taken more than one hour apart.

A) Fatigue ratings increase after main task

B) Effort discounting in early and late trials in main task

Supplementary Figure 2. (A) Fatigue ratings taken before and after the main task of the fMRI study. Each dot represents one subject and error bars reflect SEM. Fatigue was higher after the main task than before. (B) Shift in choices award from higher effort lower reward options in last 27 trials compared to first 27 trials. Shift does not occur for lowest effort highest reward offer, consistent with a shift in valuation not more random behaviour.

Another potential concern is floor/ceiling effects. It's unclear how much intra-individual variability there was in choices, which could impact interpretability of fatigue-brain relationships. Based on figure 2D, it appears that the effortful option was chosen a very high percentage of the time. At the individual level, if someone chose almost all effortful options, then we might infer that they simply did not find the task very fatiguing, in which case it become less clear how to interpret an association between the RF or UF regressor and neural activity. This could significantly influence power if the effective sample size (subjects contributing meaningful variability in choice behavior) is much lower than the actual sample size.

Response: We thank the reviewer for helping to clarify these points. Firstly, although Figure 2D highlights that lots of the offers were accepted by participants on a lot of trials, this plot was included to show that participants do rest often, but on different trials. In addition, whilst the reviewer is correct that little variability in choice behaviour could reflect that motivation is not changing for a participant over the course of the experiment, this does not mean that participants were not experiencing fatigue. Our new behavioural dataset shows that all participants show an increase in fatigue across the experiment, almost all increase their fatigue rating after a high effort. The most parsimonious explanation is therefore that some people experience fatigue but are able to persist, whereas others are less capable of enduring when experiencing similar levels of exhaustion. Thus, most participants in the fMRI study chose to rest on multiple trials, and those that only rested on a small number of trials may still have had latent sensations of fatigue. We now outline this in the revised manuscript and include the proportion of choices to rest in a histogram – supplementary Figure 4a.

Page 44

Although in this study we cannot fully rule out the possibility that this effect is simply due to time-on-task or boredom effects, we are able to show that it affects both effort-based decisions and self-reported fatigue.

Supplementary Figure 4. Supplementary results from fMRI study behaviour. (A) Histogram of proportion of trials on which participants made choices to “rest” out of 210 trials. There is considerable variability in choices both within and between many participants. (B) Exceedance probabilities for the models fitted to the choice data. Exceedance probabilities for models fitted to fMRI behavioural data. Y-axis reflects the probability of being the most frequently observed model in the population. The full model (5) is the most frequently best fitting model in the population. (C) Model parameters for each participant (green dot) and average across participants for the discounting parameter fitted to the pre-task (left), the two recoverable parameters (middle) and the unrecoverable parameter (right). Error bars represent SEM.

I appreciate the authors' incorporating a control analysis of choice-difficulty analysis. The method used for estimating choice difficulty is sound, but is susceptible to limitations for participants with highly stable choice preferences (one may agonize over a decision while still arriving to a choice consistent with model predictions). This can lead to a dramatically different scaling of trial-wise difficulty values across subjects. The authors appear to have addressed this issue by averaging difficulty values across participants, but I'm not sure this makes sense. Neural activity for the "average" choice difficulty for a particular trial is not necessarily reflective of individual differences. This may partly explain the null effects for this analysis.

Response: The reviewer raises an important point that there are multiple ways to estimate proxies of choice difficulty (Kolling et al., 2016). We appreciate that there is a limitation with the approach we have taken, that individual differences in choice difficulty may not have been taken in to account. However, the alternative approach – to use individual estimates – as the reviewer notes would also have significant limitations with hugely different ranges of variance between participants to be explained – which can lead to distorted results driven by small numbers of participant – or null results. Thus, in the revised manuscript we note the limitation to taking the approach we have, but also included a trial by trial RT analysis. Although RT analyses are also a limited proxy of choice difficulty, the inclusion of this additional information will allow readers to better understand what our results do and do not reflect.

Page 16

Moreover, we did not find that these regions specifically signalled variation in trial by trial reaction times ($p < 0.001$ uncorrected, supplementary table 4).

Page 35

In addition, to examine activity that covaried with reaction times, we also ran a separate GLM in which only individual trial by trial reaction times were included as a regressor (Supplementary table 4).

Supplementary Table 4

Anatomical locations in which activity significantly covaried with individual reaction times at $p < .001$, uncorrected for multiple comparisons

Anatomical area	MNI peak	No. of voxels	Z-value	Voxel $p_{\text{uncorr.}}$
Left anterior insula	-33, 17, 5	11158	7.69	< 0.001*
Brain stem	6, -22, -7	367	6.26	< 0.001*
Left inferior temporal gyrus	-45, -58, -13	338	5.32	< 0.001*
Right pallidum	18, 2, -1	82	4.31	< 0.001
Left cerebellum	-36, -52, -34	32	4.28	< 0.001
Right inferior temporal gyrus	48, -61, -13	154	4.23	< 0.001
Brain stem	6, -28, -34	68	4.17	< 0.001
Left medial orbital gyrus	-21, 38, -22	13	3.99	< 0.001
Left medial orbital gyrus	-15, 11, -19	8	3.61	< 0.001
Left thalamus	-12, -22, 14	8	3.55	< 0.001
Right frontal pole	21, 65, -1	6	3.52	< 0.001
Left hippocampus	-24, -25, -4	6	3.44	< 0.001
Right central operculum	48, -19, 17	8	3.32	< 0.001
Right inferior frontal gyrus	51, 47, -10	7	3.29	< 0.001

** indicates significance at a threshold of $p < .05$ with a whole-brain voxel-level family-wise error correction.*

It was unclear if proper control comparisons were performed for imaging results. For example, in the two RCZ regions associated with UF and RF, it would be useful to include the additional direct comparisons to confirm a double-dissociation. It could easily be the case that the area of RCZ showing association with RF is only slightly below SVC threshold for UF, and/or vice-versa, which would significantly change the interpretation of sub-regional specificity.

Response: We apologise to the reviewer that this was unclear in the original manuscript. All regions where we make a claim about a specific effect e.g. RF in a portion of the RCZ, we performed (i) a contrast to see if there was a significant effect of those variables independently of the others (e.g. does the RCZ show an effect of UF?), uncorrected threshold ($p < 0.001$) to examine whether voxels in the same region showed a significant effect of the other variables and (ii) we also performed contrasts (e.g. RF > UF) to ensure double dissociations where regions significantly signalled a variable significantly more than any other. We now highlight this strategy more clearly in the results section.

PAGE 14

To test our hypotheses, we first wanted to examine whether distinct regions signalled motivational fatigue states on different timescales. We therefore examined voxels in

which activity at the whole brain level and within our hypothesised regions of interest ([ROI] – See Methods) significantly covaried with RF or UF. Then we tested whether the same voxels significantly covaried with one parametric regressor and did not significantly covary with the others. Such an approach of examining overlap avoids the problems of double-dipping in ROI based analyses. Thus, results we report reflect a response exclusively to RF and UF.

In their justification for the UF/RF distinction, the authors note the work of Blain and colleagues, showing that greater fatigue led to more impulsivity/inconsistency in choice behavior. It might be interesting to test a similar idea in the current data, e.g., by examining comparing choice behavior in early and late trials in the current task.

Response: The reviewer raises an interesting point, regarding fatigue making participants more or less impulsive or inconsistent. This is an interesting idea, although it is hard to precisely quantify how impulsivity across all effort and reward levels would look like in this experiment, unlike the work of Blain and colleagues which is designed to address such questions about rewards and delay costs. However, one might expect that if participants were becoming more inconsistent that (i) they may become less effort sensitive across the experiment (i.e. there is a reduction in how much people take the effort level into account) and (ii) may show inconsistent choices towards the end of the experiment across all effort and reward levels. However, statistically we show that Effort and Reward still have very strong significant effects when examining only the last quarter of trials (Effort: $Z = -5.2319$; $p < .0001$; Reward: $Z = 4.3050$; $p < .0001$) and showed a significant cumulative effort X effort interaction ($p < 0.001$), with the effect of effort stronger at the end of the experiment than at the beginning. As such participants were basing their behaviour more strongly on the effort levels towards the end, which is not consistent with more random behaviour. In addition, participants are still choosing to work on almost 100% of trials for the highest reward and lowest effort in the last 27 trials of the main task (see supplementary figure 2), which is also inconsistent with more stochastic behaviour. In the revised manuscript we have highlighted these analyses and discussed their relation to the Blain et al. work.

Supplementary Results page 7

Could these results be due to participants becoming more random in the main task due to boredom or other confounding factors? We show that Effort and Reward still have very strong significant effects when examining only the last quarter of trials (Effort: $Z = -5.2319$; $p < .0001$; Reward: $Z = 4.3050$; $p < .0001$). As such participants were basing their behaviour strongly on the effort levels towards the end, which is not consistent with more random behaviour. In addition, participants were still choosing to work on almost 100% of trials for the highest reward and lowest effort in the last 27 trials of the main task (see supplementary figure 2), which is also inconsistent with fatigue causing more stochastic behaviour.

A) Fatigue ratings increase after main task

B) Effort discounting in early and late trials in main task

Supplementary Figure 2. (A) Fatigue ratings taken before and after the main task of the fMRI study. Each dot represents one subject and error bars reflect SEM. Fatigue was higher after the main task than before. (B) Shift in choices toward from higher effort lower reward options in last 27 trials compared to first 27 trials. Shift does not occur for lowest effort highest reward offer, consistent with a shift in valuation not more random behaviour.

This study unifies separate lines of research that have theorised that the effects of fatigue may occur on more than one timescale^{4,14}, and provides a formalised account of their effects on effort-based decisions. One line of research had suggested that extended periods of work lead to exhaustion that has consequences for tasks performed after the one which caused the fatigue^{11,17,35-37}. This “executive fatigue” influences activity in the MFG in tasks performed after having been exhausted, an effect exacerbated in athletes who are over-trained^{11,38}. This form of fatigue appears to be unrecoverable in the sense that simply taking short rests does not have a restorative effect. Although in this study we cannot fully rule out the possibility that this effect is simply due to time-on-task or boredom effects, we are able to show that it affects both effort-based decisions and self-reported fatigue. Moreover, we show that this longer-term effect is independent from a

short-term recoverable component and that it covaries with activity during effort-based choice in the left MFG, largely overlapping with an area which has previously been associated with subjective aversion to cognitive effort ³⁹.

Minor comments:

Please show discounting curves as well as parameter value distributions.

Response: We thank the reviewer for noting that we had not included the parameter values. We have now included those in supplementary figure 4. In regards to discounting curves, this is more challenging as of course the discounted value of a reward changes trial by trial according to our model. Discounting curves therefore become difficult to visualise clearly, but we hope the parameter values help demonstrate clearly the variability between participants in how value would change differently for different participants.

Supplementary Figure 4. Supplementary results from fMRI study behaviour. (A) Histogram of proportion of trials on which participants made choices to “rest” out of 210 trials. There is considerable variability in choices both within and between many participants. (B) Exceedance probabilities for the models fitted to the choice data. Exceedance probabilities for models fitted to fMRI behavioural data. Y-axis reflects the probability of being the most frequently observed model in the population. The full model (5) is the most frequently best fitting model in the population. (C) Model parameters for each participant (green dot) and average across participants for the discounting parameter fitted to the pre-task (left), the two recoverable parameters (middle) and the unrecoverable parameter (right). Error bars represent SEM.

It would be worth noting that the region of left MFG associated with UF appears similar to the region identified in a task focused on detecting effort aversion (Mcguire et al., PNAS, 2010). That might be worth discussing in terms of the interpretation of UF.

Response: We thank the reviewer for reminding us of this important paper. We have now included discussion of this work in relation to the UF result.

Page 19

This form of fatigue appears to be unrecoverable in the sense that simply taking short rests does not have a restorative effect. Although in this study we cannot fully rule out the possibility that this effect is simply due to time-on-task or boredom effects, we are able to show that it affects both effort-based decisions and self-reported fatigue. Moreover, we show that this longer-term effect is independent from a short-term recoverable component and that it covaries with activity during effort-based choice in the left MFG, largely overlapping with an area which has previously been associated with subjective aversion to cognitive effort⁴⁰.

It would be worth examining different striatal ROIs, including those associated with motor function. The authors may want to consider using the Choi 2011 parcellation seeds or some other functional parcellation of the striatum to interrogate its role more thoroughly.

Response: We thank the reviewer for highlighting the importance of precise VS localisation. Using the Harvard-Oxford atlas we were able to show that our peak coordinate lies within the Nucleus Accumbens. We have now included this information in the revised results.

Page 16

A t-contrast on SV revealed a significant positive relationship between the BOLD signal in the superior frontal gyrus (SFG) extending into the frontal pole (-12, 68, 17; $Z = 4.67$, $p = .03$ FWE) as well as in the ventral striatum, with the peak voxel within the nucleus accumbens of the Harvard-Oxford atlas (9, 11, -10; $Z = 4.31$, $p < .01$ svc).

For correlations between neural activity and model parameters, please perform comparisons of correlation coefficients (e.g., Steiger test or equivalent) to confirm differences.

Response: The reviewer raises an important point regarding correlations. However, to avoid double dipping our correlation analyses involved running an SPM covariate analysis and then identifying voxels within the VS that covaried across participants. As such, we do not have a correlation coefficient for non-significant correlations in the VS, as we have no voxels from which to extract a statistic. This avoids double-dipping but does mean that testing for

differences between correlations is not possible. We now note this limitation in the revised manuscript.

Page 35

To avoid double-dipping when correlating parameters with voxels which are already known to show a significant result in a non-independent analysis we performed these analyses by examining whether any voxels showed a significant effect within the ROI masks. Such an approach does have the limitation that the significance between correlations cannot be determined formally.

It is unclear why tables report uncorrected whole-brain values? It seems based on the text that these values were derived from a whole-brain corrected map.

Response: We apologise for the lack of clarity. The tables reporting uncorrected results were included for completeness and transparency, but all results in the main text about which inferences are made survived corrections for multiple comparisons.

Page 35

Full tables of results are reported at an uncorrected threshold $p < .001$ in Supplementary tables 1 to 3

REVIEWER COMMENTS

Reviewer #1 (Remarks to the Author):

The authors have adequately addressed the concerns except for one point. The behavioral analysis of participant choices was undertaken using a weighted summary statistic method. This is not state of the art and a hierarchical (frequentist or Bayesian) model should be estimated. I suspect that a Bayesian HM will be required by the community, but certainly a hierarchical model should be estimated.

Reviewer #2 (Remarks to the Author):

The authors have done consequent work to address all concerns. In particular, they have run a new behavioral experiment to show that their model nicely accounts for fatigue self-reports, in addition to choices made during MRI scanning. However, the manuscript has not changed much at a conceptual level, and I still find confusing the expression of motivational fatigue. The additional evidence brought during this revision only provides more support to the idea that what changes with repeated effort is simply fatigue, probably muscular fatigue or perhaps, as the authors suggest, cognitive fatigue. The allusions to potential changes in motivation, such as persistence in motivation (in the title), the idea that motivation is not static but dynamic (in the abstract), the "change in motivation" model and the proclaimed fluctuations in motivational fatigue (in the abstract, introduction and discussion) are misleading. The authors should make clear that the study is about how fatigue builds up and decreases willingness to make effort (by inflating effort cost in the expected value function). The motivational framing should be abandoned because it gives the wrong idea (that fatigue interacts with reward value instead of effort cost).

Besides, although I find appealing the tale of two fatigue states, I'm still afraid that the existence of unrecoverable fatigue is confounded with other factors (such as time on task) and that the evidence for recoverable fatigue signals in the brain is weak (borderline p-value in a region of interest picked among many possibilities). What could be done to back up the model with the two fatigue terms is a Bayesian comparison within a factorial model space (null model, effort cost affected by RF only, effort cost affected by UF only, effort cost affected by both fatigue terms). The issue with the present model comparison is that there is no clear winner, but this may relate to the peculiar model space that was explored.

Reviewer #3 (Remarks to the Author):

I appreciate the authors' thoughtful responses to my comments and congratulate them on an excellent piece of work.

I do have one remaining question - For model 7, were the pre-task k and temperature parameters used? Did they compare their winning model to the fit of the model with k and temperature estimated from the main tasks? If not, I think this might be important (i.e. to show that the more flexible model fits better than the original parabolic model that doesn't use the parameters from the pre-task that may not capture effort discounting as effectively in the main task).

Reviewer #1 (Remarks to the Author):

The authors have adequately addressed the concerns except for one point. The behavioral analysis of participant choices was undertaken using a weighted summary statistic method. This is not state of the art and a hierarchical (frequentist or Bayesian) model should be estimated. I suspect that a Bayesian HM will be required by the community, but certainly a hierarchical model should be estimated.

Response: We are pleased that our revisions have addressed the reviewer's comments and we thank them for helping us improve the manuscript.

We agree with the reviewer that hierarchical approaches offer some advantages over the classic approach taken to the model fitting in this manuscript. During this project we spent some time attempting to use both existing and bespoke approaches to fit these models in a hierarchical manner. However, no existing hierarchical approach was able to be adapted to accommodate unique features of our model that were implemented to capture fatigue. Specifically, because in this model we carried over parameters from the pre-task (k and β), into the models fitted to the main task, hierarchical approaches failed to converge. This was because a participant's fatigue parameters partially depended on the pre-task parameters – thus the distributions for each of the fatigue related parameters were non-independent from these other parameter – the end result being that attempts to refit participant parameters to a distribution produced worse model fits.

This approach of carrying parameters over was necessary and vital for modelling fatigue, as we needed to account for the known substantial individual differences that have been found in how willing people are to exert effort for reward when not fatigued (see Chong et al., 2017 for example). Thus, on balance, we decided that deploying approaches that have been used for model fitting (log-likelihood), model comparison (Bayesian information and Akaike information criterions) and testing for frequency in our population (exceedence probabilities) for many years and for a whole range of novel computational model developments still provided is with robust results. In future work we aim to devise a model fitting approach that will allow us to combine our fatigue modelling with the benefits of a hierarchical fitting procedure.

Reviewer #2 (Remarks to the Author):

The authors have done consequent work to address all concerns. In particular, they have run a new behavioral experiment to show that their model nicely accounts for fatigue self-reports, in addition to choices made during MRI scanning. However, the manuscript has not changed much at a conceptual level, and I still find confusing the expression of motivational fatigue. The additional evidence brought during this revision only provides more support to the idea that what changes with repeated effort is simply fatigue, probably muscular fatigue or perhaps, as the authors suggest, cognitive fatigue. The allusions to potential changes in motivation, such as persistence in motivation (in the title), the idea that motivation is not static but dynamic (in the abstract), the “change in motivation” model and the proclaimed fluctuations in motivational fatigue (in the abstract, introduction and discussion) are misleading. The authors should make clear that the study is about how fatigue builds up and decreases willingness to make effort (by inflating effort cost in the expected value function). The motivational framing should be abandoned because it gives the wrong idea (that fatigue interacts with reward value instead of effort cost).

Besides, although I find appealing the tale of two fatigue states, I’m still afraid that the existence of unrecoverable fatigue is confounded with other factors (such as time on task) and that the evidence for recoverable fatigue signals in the brain is weak (borderline p-value in a region of interest picked among many possibilities). What could be done to back up the model with the two fatigue terms is a Bayesian comparison within a factorial model space (null model, effort cost affected by RF only, effort cost affected by UF only, effort cost affected by both fatigue terms). The issue with the present model comparison is that there is no clear winner, but this may relate to the peculiar model space that was explored.

Response: We thank the reviewer for their additional time evaluating the manuscript. We agree with the reviewer’s interpretation of our results predominantly relating to fatigue. To address their comments we have now substantially changed the manuscript, removing the term motivation from the title, re-written large sections of the Introduction and Discussion to highlight that here we are referring the build up of fatigue, and how it impacts on decisions to exert effort for reward. Throughout the manuscript we now do not refer to the term “motivational fatigue”, and in all places we highlight that we are referring to the build up of fatigue and how it impacts on effort discounting. In places we have kept use of the term persist, because it directly relates to variability between people in choices the task – people continue to exert effort even as fatigue increases, and variability in ventral striatum signalling correlated with variability in the parameters from the model – suggesting that ventral striatum signalling may reflect how willing people are to persist through fatigue and to continue to exert effort. However, we fully appreciate the reviewer’s points and have made substantial changes throughout the manuscript.

In regards to their second point, we apologise that the Bayesian model comparison we had taken was perhaps not clear in the previous version. Our approach indeed included the proposed factorial structure (null models, RF only, UF only and RF + UF) and model comparison –in terms of AIC, BIC and exceedence probabilities did lead to strong evidence in favour of the same model in both experiments. However, over and above that factorial structure it was also necessary to include a second null

model given that there were two phases of the task which were modelled. In addition, we included a mathematically plausible, although theoretically unlikely, version of RF where its build up depended on one parameter which was included in two models (one with RF and one with RF and UF). However, we appreciate that the factorial nature of the modelling had not been made clear. In the revised manuscript we have changed several figures and descriptions of the models that more easily highlight the models of interest to the reviewer, and also show how model comparison (in terms of both AIC and exceedence probabilities) provides robust evidence for the full model of fatigue being the best at capturing both the effects of fatigue on choice data and ratings data (as can be seen in the revised figures). Given that this model wins, with both recoverable and unrecoverable factors necessary to explain changes in effort-based choice, and trial by trial ratings of fatigue, we believe this provides strong evidence of both factors underlying sensations of fatigue and the willingness to exert effort.

Revised text (model description – page 32)

To verify whether the three parameters used to quantify the effects of fatigue were necessary, alternative models were also fitted to participants' choices in the main task. These models fit within a factorial structure of models containing no effect of fatigue (two null models), an effect of UF only (i.e. θ being fitted), an effect of RF only (i.e. α and δ being fitted) or the full model with both RF and UF. The two null models predicted no effect of fatigue in the main task, one which used the original pre-task discounting parameter (k) and thus assumed that the willingness to exert effort stayed the same across the whole experiment, and a second where a new discounting parameter (γ) was calculated across all trials in the main task which assumed a fixed change in the willingness to work between the pre and main tasks. In addition, two further mathematically plausible, but theoretically unlikely, models were included which used only one parameter to scale the effect of effort and rest on recoverable fatigue (i.e. only α being fitted across both work and rest trials). In one of these models this fatigue was only comprised by this one parameter RF, while in a second model, fatigue comprised UF plus the one parameter RF. These two models had higher AIC values and thus worse fits than versions of the RF model including separate parameters and are thus not shown in figures. In the models including a fatigue term, initial *RF* and *UF* values were defined such that the initial total fatigue was always equal to 1.

Revised Figure 3

A) Models of effort-discounting with / without fatigue

Null models (no effects of fatigue):

model 1: Static effort discounting (No change)

$$SV_{(t)} = R_{(t)} - k * E^2_{(t)}$$

model 2: New discounting parameter (No fatigue)

$$SV_{(t)} = R_{(t)} - \gamma * E^2_{(t)}$$

Fatigue-weighted effort-discounting models:

$$SV_{(t)} = R_{(t)} - F_{(t)} * k * E^2_{(t)}$$

model 3: Unrecoverable fatigue only (UF)

$$F = UF$$

$$UF_{(t)} = UF_{(t-1)} + (\theta * E_{(t-1)})$$

model 4: Recoverable fatigue only (RF)

$$F = RF$$

$$RF_{(t)} = RF_{(t-1)} + (\alpha * E_{(t-1)}) - (\delta * T_{rest(t-1)})$$

model 5: Full model of fatigue (UF + RF)

$$F = UF + RF$$

$$RF_{(t)} = RF_{(t-1)} + (\alpha * E_{(t-1)}) - (\delta * T_{rest(t-1)})$$

$$UF_{(t)} = UF_{(t-1)} + (\theta * E_{(t-1)})$$

B) Unrecoverable and Recoverable fatigue increase the parabolic discounting of rewards by effort

D) Logistic Regression: Fatigue values in model predict choices to work

Figure 3. Modelling the fatigue-weighted subjective value of effort and reward. (A) List of models compared. All models assume that rewards (R) increase subjective value (SV), effort (E) decreases subjective value, and people discount the rewards by effort idiosyncratically – modelled with a discount parameter ‘k’ fitted to the pre-task. Two null models assume that the willingness to exert effort for reward remains static across the trials of the main task, either with the same discounting parameter as for the pre-task (k; model 1) or with a new discounting parameter (γ; model 2). Models 3 to 5 capture changes in effort discounting due to fatigue. The full model (model 5) assumes that exerting effort increases recoverable fatigue (RF), but time (T_{rest}) spent resting decreases it. Both of these are scaled for each participant by two corresponding free parameters that define a person’s short-term fatigability (α, δ). Unrecoverable fatigue also increases through exerted efforts, but never declines here. This is also weighted by an idiosyncratic free parameter (θ), which defines long-term fatigability. These are summed to create F, which then serves to increase the discounting of rewards by effort as they increase. Models 3 and 4 include only the effects of UF or RF. (B) Schematic representation for how F (both UF and RF combined) affect value and choices to work or rest, with greater discounting of rewards by effort as fatigue increases. (C) Model comparison highlights that the full model is a better predictor of choice data than the simpler models in terms of AIC (left) and in exceedance probabilities (right), highlighting that both RF and UF are necessary to understand the willingness to exert effort, Star indicates winning model. (D) Furthermore, the model-estimated RF and UF – from model 5 – significantly interacted with effort to predict choice behaviour in a logistic regression. The asterisks show significant t-scores (p < .001) and the error bars represent SEM.

C) Model comparison – necessity for all parameters in full model

A) Model comparison reveals best fit by full model

B) Full model is most frequently observed in the population

C) Models tested

model 1: Unrecoverable fatigue only (UF)

$$F = UF$$

$$UF_{(t)} = UF_{(t-1)} + (\theta * E_{(t)})$$

model 2: Recoverable fatigue only (RF)

$$F = RF$$

$$RF_{(t)} = RF_{(t-1)} + (\alpha * E_{(t)}) - (\delta * T_{rest(t)})$$

Model 3: Full model of fatigue (UF + RF)

$$F = UF + RF$$

$$RF_{(t)} = RF_{(t-1)} + (\alpha * E_{(t)}) - (\delta * T_{rest(t)})$$

$$UF_{(t)} = UF_{(t-1)} + (\theta * E_{(t)})$$

Supplementary Figure 7. Computational modelling results for behavioural rating study. (A) Model comparison results in AIC for the models predicting changes in fatigue ratings across the experiment. The full model is the best fit to the data when punishing for the number of parameters. (B) Exceedance probabilities for the models fitted to the ratings data. Y-axis reflects the probability of being the most frequently observed model in the population. Model 3 is the winning “full” model of fatigue containing separate RF and UF components. (C) Models compared. All models predicted changes in fatigue ratings. The best fitting model predicted changes in fatigue that were partially recoverable - increase through effort, decrease through rest – but also contained a long-term unrecoverable component.

Reviewer #3 (Remarks to the Author):

I appreciate the authors' thoughtful responses to my comments and congratulate them on an excellent piece of work.

I do have one remaining question - For model 7, were the pre-task k and temperature parameters used? Did they compare their winning model to the fit of the model with k and temperature estimated from the main tasks? If not, I think this might be important (i.e. to show that the more flexible model fits better than the original parabolic model that doesn't use the parameters from the pre-task that may not capture effort discounting as effectively in the main task).

Response: We thank the reviewer for their congratulations and kind comments on the work and revisions. The reviewer raises an important point that an additional model with free K and beta parameters in the softmax would be informative. They were correct in that model 7 did use the fixed parameters, however, the original model 6 (model 2 in the latest revision) did use a refitted parameter (γ) in replacement of the pre-task discounting parameter (k) to address exactly the question by the reviewer. There was little evidence that this model was able to capture choice behaviour effectively, with an exceedence probability close to 0. In the revised manuscript we have modified the descriptions of the models and how they are presented in figures to make this clearer to address the reviewers' comment and those of reviewer 2. We thank you for helping us improve our work.

REVIEWERS' COMMENTS

Reviewer #2 (Remarks to the Author):

The authors should be praised for their careful revisions.
I see no good reason to delay publication any longer.

Reviewer #3 (Remarks to the Author):

The authors have addressed my concerns.

REVIEWERS' COMMENTS

Reviewer #2 (Remarks to the Author):

The authors should be praised for their careful revisions.
I see no good reason to delay publication any longer.

Authors Response: We thank the reviewer for their helpful comments and kind words.

Reviewer #3 (Remarks to the Author):

The authors have addressed my concerns.

Authors Response: We thank the reviewer for their helpful comments.